**Methane distribution and oxidation around the Lena delta in summer 2013**

Ingeborg Bussmann, Steffen Hackbusch, Patrick Schaal, Antje Wichels

Alfred Wegener Institute for Polar and Marine Research, Marine Station Helgoland, Kupromenade 201, 27498 Helgoland, Germany

*Correspondence to*: Ingeborg Bussmann (Ingeborg.bussmann@awi.de)

**Abstract.** The Lena River is one of the largest Russian rivers draining into the Laptev Sea. The predicted increases in global temperatures are expected to cause the permafrost areas surrounding the Lena delta to melt at increasing rates. This melting will result in high amounts of methane reaching the waters of the Lena and the adjacent Laptev Sea. The only biological sink that can lower methane concentrations within this system is methane oxidation by methanotrophic bacteria. However, the polar estuary of the Lena River, due to its strong fluctuations in salinity and temperature, is a challenging environment for bacteria. We determined the activity and abundance of aerobic methanotrophic bacteria by a tracer method and by the quantitative polymerase chain reaction. We described the methanotrophic population with a molecular fingerprinting method (monooxygenase intergenic spacer analysis), as well as the methane distribution (via a head-space method) and other abiotic parameters, in the Lena delta in September 2013.

The median methane concentrations were 22 nmol $L^{-1}$ for riverine water (salinity (S) <5), 19 nmol $L^{-1}$ for mixed water (5 < S < 20) and 28 nmol $L^{-1}$ for polar water (S > 20). The Lena River was not the source of methane in surface water, and the methane concentrations of the bottom water were mainly influenced by the methane concentration in surface sediments. However, the bacterial populations of the riverine and polar waters showed similar methane oxidation rates (0.419 and 0.400 nmol $L^{-1} d^{-1}$), despite a higher relative abundance of methanotrophs and a higher estimated diversity in the riverine water than in the polar water. The methane turnover times ranged from 167 d in mixed water and 91 d in riverine water to only 36 d in polar water. The environmental parameters influencing the methane oxidation rate and the methanotrophic population also differed between the water masses. We postulate the presence of a riverine methanotrophic population that is limited by sub-optimal temperatures and substrate concentrations and a polar methanotrophic population that is well adapted to the cold and methane-poor polar environment but limited by a lack of nitrogen. The diffusive methane flux into the atmosphere ranged from 4 to 163 $\mu$mol $m^2 d^{-1}$ (median 24). The diffusive methane flux accounted for a loss of 8% of the total methane inventory of the investigated area, whereas the methanotrophic bacteria consumed only 1% of this methane inventory. Our results underscore the importance of measuring the methane oxidation activities in polar estuaries, and they indicate a population-level differentiation between riverine and polar water methanotrophs.

## 1. Introduction

Methane is an important greenhouse gas and concerted efforts are ongoing to assess its different sinks and sources. Overall, about two-thirds of methane emissions are caused by human activities; the remaining third arises from natural sources (Kirschke et al., 2013). Methane sources and sinks also vary with latitude (Saunois et al., 2016); for example, methane sources at polar latitudes include wetlands, natural gas wells and pipelines, thawing permafrost, and methane hydrate associated with decaying offshore permafrost (Nisbet et al., 2014). The top-down and bottom-up estimates of methane from these various sources also show a divergence, so more data are needed, but the measurement network that focuses on methane concentrations and isotopes is rather sparse (Nisbet et al., 2014). Better measurements, both spatial and temporal, are essential for identifying and quantifying methane sources.

One poorly studied area is the Arctic Ocean, the intercontinental sea that is surrounded by the landmasses of U.S.A. (Alaska), Canada, Greenland, Norway, Iceland and Russia (Siberia). This ocean represents only about 1% of the global ocean volume, but it receives about 10% of all global river runoff (Lammers et al., 2001). It has a deep central basin and is characterised by extensive shallow shelf areas, including the Barents, Kara, Laptev, East Siberian, Chukchi and Beaufort Seas. Methane sources in these arctic areas can include the thawing methane hydrates off the coast of Svalbard (Westbrook et al., 2009) and ebullition of methane from diverse geologic sources (Mau et al., 2017; Shakhova et al., 2014). In addition, the extensive shallow-water areas of the Arctic continental shelf are underlain by permafrost, which was formed under terrestrial conditions and subsequently submerged by post-glacial rises in sea level. Methane trapped within this permafrost, as well as below its base (Rachold et al., 2007), can serve as yet another source of this greenhouse gas.

The fate of released methane depends on several factors. When methane leaves the sediment (either by diffusion or by ebullition) at water depths > 200 m, most of it is dissolved into the water below the thermocline and does not reach surface waters or the atmosphere (Gentz et al., 2013; Myhre et al., 2016). However, at shallow water depths, most of the methane released by ebullition does not dissolve in the water but instead is released into the atmosphere. For lakes, ebullition is estimated to contribute 18–22% of the total methane emission (Del Sontro et al. 2016).

Methane that does dissolve in the water can be oxidised by methane oxidising bacteria (MOBs). These microorganisms can convert methane to $CO_2$ and water, thereby considerably reducing the greenhouse effect (Murrell and Jetten, 2009). Methane oxidation in the water column therefore represents an important methane sink before its release from the aquatic system into the atmosphere. The amount of methane consumed by this microbial filter depends on the abundance of MOBs and the water current patterns (Steinle et al., 2015). The efficiency of MOBs is determined mostly by methane concentrations and temperature (Lofton et al., 2014), but not much is known about the abundance and population structure of marine MOBs from polar habitats.

The area of the Laptev and East Siberian Seas has been a scientific focus of polar methane studies. The partial thawing of the permafrost on the shallow East Siberian Arctic Shelf is viewed as the source of the very high dissolved methane concentrations found in the water column (> 500 nmol L$^{-1}$) and of the elevated methane concentrations measured in the atmosphere (Shakhova et al., 2014). Other authors have shown that methane released from thawing permafrost in the Laptev Sea region is efficiently oxidised by microorganisms in the overlying unfrozen sediments so that methane concentrations in the water column are close to the normal

background levels (Overduin et al., 2015). High-resolution, simultaneous measurements of methane in the atmosphere above and in surface waters of the Laptev and East Siberian Seas have revealed that the sea–air methane flux is dominated by diffusive fluxes, not bubble fluxes (Thornton et al., 2016).

The aim of the present study was to obtain an overview of the methane distribution in the northern parts of the Lena delta and to gain the first key insights into the role of MOBs in the methane cycle occurring in this area. An additional aim was to assess which environmental factors determine methane distribution and oxidation in this delta.

## 2.   Materials and Methods

### 2.1   Study site

The Lena Expedition was conducted in late summer, 1–7 September, 2013, on board the Russian research vessel RV *Dalnie Zelentsy* of the Murmansk Marine Biological Institute, in the areas surrounding the Lena River delta of the Laptev Sea, Siberia. Four transects around the Lena delta were investigated (Figure 1). Transect 1 started near the Bykovski peninsula and headed towards the northeast; this was the same transect studied in 2010 (Bussmann, 2013b). Transect 4 was located near the mouth of the Trofimovskaya Channel, and Transect 6 was located at the northern point of the delta.

Hydrography (temperature, salinity, currents) and water chemistry (dissolved organic carbon [DOC], pH, oxygen, total dissolved nitrogen [TDN]) were evaluated as described previously (Gonçalves-Araujo et al., 2015; Dubinenkov et al., 2015). Water samples were taken using Niskin bottles at the surface and at discrete depths chosen based on salinity profiles. Samples for methane analyses were taken from surface and bottom waters and at the pycnoclines at the deeper stations. The sediment surface was sampled with a grab sampler.

Using a modification of the classification system of Caspers (1959), we classified the water masses as riverine water (salinity (S) < 5), mixed water (5 < S < 20) and polar water (S > 20).

### 2.2   Water sampling and gas analysis

Duplicate serum bottles (120 mL) were filled from the water sampler using thin silicon tubing. The bottles were flushed extensively with sample water (to ensure no contact with the atmosphere) and finally closed with butyl rubber stoppers; excess water could escape via a needle in the stopper. Samples were poisoned with 0.3 mL of 25% $H_2SO_4$, stored upside down at temperatures < 15°C, and analysed after 4 months. Glass bottles and butyl stoppers are relatively methane tight and acidification of water samples results in good long-term sample preservation (Magen et al., 2014; Taipale and Sonninen, 2009). However, we cannot exclude the possibility that some methane was lost from the samples. In the home laboratory, 20 mL of nitrogen was added to extract the methane from the water phase, and excess water was allowed to escape via a needle. The samples were vigorously shaken and equilibrated for at least two hours. The volumes of the water and gas phases were determined gravimetrically.

For sediment samples, 3 mL of surface sediment was transferred into 12 mL glass ampoules using cut off syringes. The samples were poisoned with 2 mL NaOH (1 mol $L^{-1}$) and sealed with butyl rubber stoppers.

Headspace methane concentrations were analysed in the home laboratory with a gas chromatograph (GC 2014, Shimadzu) equipped with a flame ionisation detector and a molecular sieve column (Hay Sep N, 80/100, Alltech). The temperatures of the oven, the injector and detector were 40, 120 and 160 °C, respectively. The carrier gas ($N_2$) flow was 20 mL min$^{-1}$, with 40 mL min$^{-1}$ $H_2$ and 400 mL min$^{-1}$ synthetic air. Gas standards (Air Liquide) with methane concentrations of 10 and 100 ppm were used for calibration. The calculation of the methane concentration was performed according to Magen et al. (2014), taking into account the different methane solubilities at the wide range of salinities (1–33). The precision of the calibration line was $r^2 = 0.99$ and the reproducibility of the samples was 7%. The methane-related data set is available at www.pangaea.de, doi:10.1594/PANGAEA.868494, 2016.

### 2.3 Determination of the methane oxidation rate (MOX)

The MOX was determined as described previously (Bussmann et al., 2015). After filling triplicate sample bottles and one control bottle, a diluted tracer (0.1 mL of $^3$H-CH$_4$, American Radiolabeled Chemicals) was added to the samples (2 kBq mL$^{-1}$). The samples were shaken vigorously and incubated for 24 hours in the dark at near in situ temperatures (approximately 4–10°C). After incubation, methane oxidation was stopped by adding 0.3 mL of 25% $H_2SO_4$. (Controls were stopped before the addition of the tracer.) The principle of the MOX estimation is the comparison between the total amount of radioactivity added to the water sample and the radioactive water that was produced due to oxidation of the tritiated methane. The ratio between these values, corrected for the incubation time, is the fractional turnover rate (k'; d$^{-1}$). The in situ MOX (nmol L$^{-1}$ d$^{-1}$) is then obtained by multiplying k' with the in situ methane concentration. We also calculated the turnover time (1 /k') (i.e. the time it would take to oxidise all the methane at a given MOX, assuming that methane oxidation is a first-order reaction). The total radioactivity of the sample and the radioactivity of the tritiated water were determined by mixing 4 mL aliquots of water with 10 mL of scintillation cocktail (Ultima Gold LLT, Perkin Elmer) and analysing with a liquid scintillation counter (Beckman LS 6500). The limit of detection was calculated as described previously (Bussmann et al., 2015) and was determined to be 0.028 nmol L$^{-1}$ d$^{-1}$ for this data set.

### 2.4 PCR amplification of methane monooxygenase genes

Samples (250 mL) from surface and bottom water were filtered through 0.2 $\mu$m cellulose acetate filters (Sartorius) and stored frozen until further processing. High molecular weight DNA was extracted following the protocol of the PowerWater® DNA Isolation Kit (MoBio). DNA concentrations were determined photometrically (TECAN infinite200). Each DNA sample was checked for the presence of methanotrophic DNA with the primers wcpmoA189f / wcpmoA661r, as water-column–specific primers (Tavormina et al., 2008). Each PCR reaction (30 $\mu$L) contained 2 U of Taq Polymerase (5 Prime), 3 $\mu$L PCR Buffer (10×), 6 $\mu$L PCR Master Enhancer (5×), 200 $\mu$M dNTP Mix (10 mM Promega), 0.6 $\mu$M of each primer, and 10 ng of DNA template. Initial denaturation at 92 °C for 180 s was followed by 30 cycles of denaturation at 92 °C for 30 s, annealing at 59 °C for 60 s and elongation at 72 °C for 30 s. The final elongation step was at 68 °C for 300 s. Successful amplification was confirmed by gel electrophoresis on a 1.5% (w/v) agarose gel.

### 2.5 Quantitative PCR (qPCR) of methane monooxygenase genes

Extracted DNA from each sample was amplified by qPCR using a LightCycler R 480 (Roche, Germany) and master mixes from the company (Roche, Germany). Each sample was measured in triplicate.

A pure culture of *Methylobacter luteus* (NCIMB 11914) was used to construct standard curves for the total *pmoA* gene. The *M. luteus* cultures were stained and cell numbers were determined with a microscope. DNA was extracted and quantified using a TECAN infinite M200 spectrophotometer (TECAN, Switzerland). A serial dilution of DNA (equivalent to $10$–$10^6$ cells mL$^{-1}$) was used to construct standard curves. Correlation coefficients of standard curves were > 0.98. The relative abundance was calculated as the percentage of MOB-DNA in the total extracted DNA.

The qPCR reaction mix (20 $\mu$L) contained 10 $\mu$L Master Mix (2 × LightCycler® 480 kit hot-start SYBR Green I Master, Roche, Germany), 10 mM of each PCR-primer (as described above) and 5 $\mu$L template DNA. The amplification was performed with an initial denaturation step at 95 °C for 5 min, followed by 45 cycles of denaturation at 95 °C for 10 s, annealing at 59 °C for 60 s and extension at 72 °C for 30 s. Fluorescence data were acquired during an additional temperature step (60 s at 65 °C).

### 2.6 Methane monooxygenase intergenic spacer analysis (MISA)

All samples showing *pmoA* genes were analysed with MISA to differentiate the methanotrophic populations and to describe their estimated diversity by analysing the differences in the composition of methane monooxygenase genes with regard to their geographical distribution (Tavormina et al., 2010).

The PCR master mix (20 µl) contained 200 $\mu$M dNTPs, (Promega), 2 U Taq DNA polymerase (5 Prime), 2 $\mu$L PCR Buffer (10x), 4 $\mu$L PCR Master Enhancer (5 ×) and 15 ng target DNA. Two PCR runs were carried out with a MasterCycler gradient (Eppendorf, Germany) modified after Tavormina et al. (2010) using two sets of primers (Thermo Fisher Scientific GmbH, Germany): The *pmoA* sequences were enriched from bulk environmental DNA using the primers spacer_pmoC599f (5'-AAYGARTGGGGHCAYRCBTTC), spacer_pmoA192r (5'-TCDGMCCARAARTCCCARTC). A second round of semi-nested amplification was performed using the primers spacer_pmoC626_IRD (5'-RCBTTCTGGHTBATGGAAGA) and spacer_pmoA189r (5'-CCARAARTCCCARTCNCC) and the purified PCR product from the first PCR as the template. Primer spacer_pmoC626_IRD is labelled with an infrared dye (Dy 682 nm) for the detection of amplified products using a Licor DNA Analyser 4300 system (Licor, Germany). Primers are modified versions of MISA primers, as reported previously (Tavormina et al., 2010). Modifications used in the current work increased amplicon strength and recovery of diverged lineages (Tavormina, pers. comm.). In detail, in the first PCR, an initial denaturation at 94 °C for 180 s was followed by 30 cycles of denaturation at 94 °C for 30 s, annealing at 52 °C for 60 s and elongation at 72 °C for 30 s. The final elongation step was at 72 °C for 300 s. In the second PCR, 2 $\mu$L of purified PCR product from the first PCR was used for amplification with modified and labelled primers (see above). The PCR program was modified as follows: initial denaturation at 94 °C for 180 s was followed by 5 cycles of denaturation at 94 °C for 30 s, annealing at 52 °C for 60 s, elongation at 72 °C for 30 s and 25 cycles with an annealing temperature of 48 °C.

Amplified samples were separated on polyacrylamide gels using a DNA Analyser 4300 (Licor, Germany). Running conditions on a 6.5% polyacrylamide gel (Lonza, Switzerland, 25 cm length, 0.25 mm thickness) were 1500 Volt, 40 mA, 40 W for 3.30 h at 45 °C. A 50–700 bp sizing standard (IRDye 700, Licor, Germany) was applied to the gel. For the analysis of the MISA fingerprints (Bionumerics 7.0, Applied Maths, Belgium), size fragments of 350 to 700 bp were included (Schaal, 2016). Binning to band classes was performed with a position tolerance setting of 1.88%. Each band class is referred to as a MISA operational taxonomic unit (MISA-OTU). Band patterns of the MISA-OTUs were translated to binary data reflecting the presence or absence of the respective OTU. The estimated diversity of MOBs was defined as the number of OTUs per station.

## 2.7    Calculation of the diffusive methane flux

The gas exchange across an air–water interface can be described in general by the following function (Wanninkhof et al., 2009):

$$F = k_{CH4} * (c_m - c_{equ})$$

where F is the rate of gas flux per unit area (mol m$^{-2}$ d$^{-1}$), $c_m$ is the methane concentration measured in surface water and $c_{equ}$ is the atmospheric gas equilibrium concentration (Wiesenburg and Guinasso, 1979). Data on the atmospheric methane concentration were obtained from the meteorological station in Tiksi via NOAA, Earth System Research Laboratory, Global Monitoring Division (http://www.esrl.noaa.gov/gmd/dv/iadv/). The gas exchange coefficient (k) is a function of water surface agitation. The k value in oceans and estuaries is determined mostly by wind speed, whereas water velocity dominates in rivers (Alin et al., 2011). The determination of k is very important for the calculation of the sea–air flux. We decided to calculate $k_{600}$ in the Laptev Sea according to the following equation, obtained for coastal seas (Nightingale et al., 2000):

$$k_{600} = 0.333 \, U_{10} + 0.222 \, U_{10}^2$$

Wind data ($U_{10}$) were obtained for Tiksi from the 'Archive of Tiksi for Standard Meteorological Observations, 2016'. The median wind speed of each day was used for the flux calculation. The calculated $k_{600}$ (value for $CO_2$ at 20°C) was converted to $k_{CH4}$ (Striegl et al., 2012), where Schmidt numbers (Sc) are determined by water temperature and salinity (Wanninkhof, 2014):

$$k_{CH4} / k_{600} = (Sc_{CH4} / Sc_{CO2})^{0.5}$$

The role of methane oxidation and the diffusive methane flux for the methane inventory in the Lena delta were estimated using the following calculations. Two rectangles, which are bordered by the most southern, northern, eastern and western stations, gave a good estimation of investigated area (Figure 1). The median depth from the stations within each of these rectangles was 13 m. Based on the longitude and latitude of the rectangle, we calculated the area ($1.02 \times 10^{10}$ m$^2$ and $2.01 \times 10^{10}$ m$^2$) and then the volume of each rectangle ($1.3 \times 10^{11}$ m$^3$ and $2.5 \times 10^{11}$ m$^3$). Using the median methane concentration and median MOX of all stations within each rectangle, we calculated the total methane inventory of the investigated areas (in mol, as the sum of both rectangles), as well as the total methane oxidation rate (mol / d). The total diffusive flux (in mol / d) of the region was obtained by multiplying the median diffusive flux for all stations by the total area.

### 2.8 Statistical analysis

We tested for differences between the different water masses by applying a one-way analysis of variance (ANOVA) with log transformed data (Kaleidagraph 4.5). We tested for differences between different groups using the non-parametric Wilcoxon Rank Sign Test or Kruskal Wallis test (Kaleidagraph 4.5). The linear correlation analyses were also performed with log transformed data and Kaleidagraph 4.5. Outliers were defined as points whose values are greater than UQ + 1.5 * IQD or less than LQ – 1.5 * IQD; with UQ = upper quartile, LQ = lower quartile and IQD = interquartile distance (Kaleidagraph 4.5). Outliers were excluded from further statistical analyses.

## 3. Results

### 3.1 Hydrography

We grouped our sampling stations into riverine water with a salinity < 5. In this water mass, the median salinity was 2.45, ranging from 0.8–4.8, and the median temperature was 9.8 °C, ranging from 7.3–11.4 °C. In the mixed water, the median salinity was 11.4, ranging from 5–19.7, and the median temperature was 6.4 °C, ranging from 2.5–8.8 °C. In the polar water, the median salinity was 27.2, ranging from 21.5–33.2, and the median temperature was 3.0 °C, ranging from 1.8–6.2 °C. In September 2013, we observed a sharp stratification, with a warm freshwater layer at the surface (0–5 m) and a mixed water layer immediately below that. Water at depths greater than approximately10 m consisted of cold and saline water (= polar water). This sharp stratification is illustrated by the salinity distribution of Transect 1 shown in Figure 2a. The freshwater plume was most pronounced in Transects 4 and 5 and extended far to the north (Appendix Figure A1). In Transect 6, only the first near-shore station had riverine water; the stations farther off shore were characterised by polar waters.

### 3.2 Methane concentrations

The methane concentrations around the Lena delta were elevated near the shore and decreased with distance from the shore (Figure 3). The decrease off the coast was most distinct for Transects 1 and 4, which also had the maximal methane concentrations (218 nmol $L^{-1}$). At station TIII-13 04, we also observed high methane concentrations at the surface (212 nmol $L^{-1}$; Figure 3). By contrast, methane concentrations were distributed rather uniformly in the northern Transect 6. No clear pattern was observed in the depth distribution of methane (Figure 2b). The methane concentrations of the sediment surface ranged from 430 nmol $L^{-1}$ at the eastern station of Transect 4 to 5380 nmol $L^{-1}$ at the beginning of Transect 1 (the overall median concentration was 2070 nmol $L^{-1}$).

We observed significantly different methane concentrations in the riverine, mixed and polar water masses, with medians of 22, 19 and 26 nmol $L^{-1}$, respectively (p = 0.03; Table 1). Therefore, we conducted separate linear correlation analyses for each water mass.

In riverine water, the methane concentration was significantly positively correlated with temperature ($r^2 = 0.38$, Table 2) and negatively correlated with the oxygen concentration ($r^2 = 0.73$). In mixed water, we found a weak

but significant correlation between methane and TDN ($r^2 = 0.27$, Table 2). In polar water, the methane concentration of the water column was significantly correlated with the methane concentration in the surface sediment ($r^2 = 0.33$). The influence of the sediment methane concentration on the water column concentration was even more pronounced when taking all bottom water samples (= polar water + one mixed water + one riverine sample) and excluding the very high methane concentrations detected at station TIII-1304. These two modifications gave a much stronger correlation ($r^2 = 0.62$, n= 33, Figure 4).

### 3.3.  Methane oxidation rate (MOX) and fractional turnover (k')

The MOX ranged from below the detection limit (< 0.028 nmol $L^{-1}d^{-1}$, in 8.7% of the data) up to 5.7 nmol $L^{-1}d^{-1}$. In riverine and polar water, methane oxidation was rather high (median of 0.419 and 0.400 nmol $L^{-1}d^{-1}$), when compared to the low rates observed in mixed water (median of 0.089 nmol $L^{-1}d^{-1}$, Table 1). On a spatial range, we observed slightly elevated rates near the coast, at the beginning of Transects 1 and 4 (Figure 5a). In the bottom waters, elevated values were also observed near the coast, at the beginning of Transects 4 and 5.

In the riverine water, the MOX showed a significant positive correlation with temperature ($r^2 = 0.77$, Table 3). In mixed water, none of the measured parameters showed statistically significant correlations. In polar water, TDN explained 31% of the observed MOX variability, although at a low level of significance (p<0.1). In all water masses, MOX was influenced by the methane concentration, but the influence was strongest in riverine water ($r^2 = 0.98$) and lower in mixed and polar water ($r^2 = 0.80$ and 0.56 respectively, Table 3). However, as MOX is calculated based on the methane concentration, this correlation has to be regarded with caution.

The fractional turnover (k') is a measure of the relative activity of the MOBs, and it is independent of the methane concentration. We observed significantly different k' values in riverine, mixed and polar water (medians of 0.011, 0.006 and 0.028 $d^{-1}$, respectively, Table 3), with the highest k' in polar water. Temperature was the most important parameter for the k' in riverine water ($r^2 = 0.84$). In mixed water, salinity and TDN correlated with k' ($r^2 = 0.46$ and 0.37, respectively). In polar water, none of our parameters correlated with k' (Table 3).

### 3.4  Relative abundance of methane oxidising bacteria (MOBs)

The abundance of MOBs can be expressed as cell numbers or as relative abundance. Cell numbers ranged from $4.0 \times 10^4$ to $4.6 \times 10^5$ cells $L^{-1}$, except at station T1-1302, which had very high numbers of $2 \times 10^6$ to $3 \times 10^6$ cells $L^{-1}$. The relative abundance ranged from 0.05 to 0.47%, except for the high values from station T1-1302, at 1.69 and 2.63% (surface and bottom, respectively, Figure 6). These high values could not be explained by any environmental or methane-related parameters. In addition, they were statistical outliers and were excluded from further analysis. The detection limit was $3.2 \times 10^4$ cells per L, and about one quarter of the samples were below this limit.

The relative abundance of MOBs was significantly different between riverine, mixed and polar waters (Table 1). The highest relative abundance was found in riverine water, followed by mixed water and then polar water (median values of 0.81, 0.19 and 0.03% respectively).

For further analysis, we excluded the outliers that had very high values. Since the total number of data points was small (n = 18), we performed a linear regression analysis with all values (no separation of the different

water masses). None of the methane-related parameters (methane concentration, MOX and k') could explain the observed relative abundance of MOBs. However, the relative abundance of MOBs was significantly and positively correlated with DOC ($r^2 = 0.52$; p = 0.0002) and temperature ($r^2 = 0.41$; p = 0.0002) and negatively correlated with salinity ($r^2 = 0.47$; p <0.0001). The estimated diversity (OTUs per station) also showed a weak but significant correlation with relative abundance ($r^2 = 0.20$; p = 0.04). Similar results were obtained when using the cell numbers as a dependent parameter.

## 3.5    Methanotrophic population

The MISA fingerprinting method allowed the detection of nine different OTUs, which we named according to their PCR fragment length (size in bp). Of these, two OTUs (420 and 506) were observed at all stations and at all depths. Their occurrence pattern therefore could not provide any ecological information, so they were excluded from further analysis. The estimated diversity of MOBs, as the number of OTUs per station, differed significantly between riverine, mixed and polar waters, with four, three and two OTUs per station, respectively (Kruskal Wallis test, p = 0.02, Table 4). The Kruskal-Wallis test was applied for each OTU (presence / absence data) to analyse the association with the three water masses. OTU-557 showed a clear association with polar water (p =0.06), while OTU-460 and OTU-398 were absent from polar water. OTU-535 showed a significant association with river and mixed water (p=0.02), as did OTU-362 (although this association was not statistically significant). OTU-485 and OTU-445 showed no clear associations. With respect to the PCR fragment size, some of the OTUs have been described previously (Tavormina et al., 2010); thus, OTU-535 could be assigned to Group Z, OTU-485 to *Methylococcus capsulatus* and *Methylohalobius crimeensis* and OTU-445 to OPU-1 (Table 4).

## 3.6    Diffusive methane flux

Calculation of the diffusive flux of methane requires information on the atmospheric methane concentration as well as the wind speed for the respective dates, as outlined in Sect. 2.7. The atmospheric methane concentration ranged from 1.896 to 1.911 ppm $CH_4$. The wind speed in September 2013 was rather low, at 4.2 ± 2.2 m/s. The calculated values for $k_{600}$ ranged from 0.37 to 3.17 m d$^{-1}$, with a median of 1.05 m d$^{-1}$, while $k_{CH4}$ ranged from 0.52 to 4.51 m d$^{-1}$, with a median of 1.43 m d$^{-1}$.

The diffusive flux of methane into the atmosphere was rather low for Transects 1, 5 and 6, with median values of 31, 8 and 13 $\mu$mol m$^{-2}$ d$^{-1}$, respectively, compared to a median flux of 163 $\mu$mol m$^{-2}$ d$^{-1}$ for Transect 4. The highest flux was observed at the near shore stations of Transect 4, at 478 and 593 $\mu$mol m$^{-2}$ d$^{-1}$; this was mainly due to higher methane concentrations (118 and 151 nmol L$^{-1}$) and higher wind speeds on the sampling day.

Our cruise covered a total area of 3051 km$^2$ (Figure 1), with an inventory of 10,161 kmol methane. Based on our estimations, about 822 kmol per day (the median value of all stations) diffused into the atmosphere, while 118 kmol per day (the median value of all stations) were oxidised. Thus, about 8% of the total methane inventory leaves the aquatic system via diffusion, whereas only 1% is oxidised each day.

## 4. Discussion

### 4.1 Methane concentrations

In the coastal area of the Laptev Sea, we observed rather low methane concentrations (overall median 25 nmol $L^{-1}$, ranging from 10 to 218 nmol $L^{-1}$). Transect 1 was located at the same latitude and longitude as in our expedition in 2010 (Bussmann, 2013a). Near the shore, methane concentrations were slightly higher in 2013, but there was no significant difference overall (Wilcoxon Rank Sign Test for paired data; n = 18, p = 0.84). In the same study area and in the summer of 2014, other authors reported a range of 10 to 100 nmol $L^{-1}$ (Sapart et al. (2017), as estimated from Figure 2 of that paper). Two other arctic estuaries, the Ob and the Yenisei, showed similarly low concentrations, at $18 \pm 16$ nmol $L^{-1}$ (Savvichev et al., 2010) and approximately 30 nmol $L^{-1}$ (Kodina et al., 2008), respectively. Near the Alaskan coast, maximal concentrations of 50 nmol $L^{-1}$ have been reported for stations with $\leq 20$ m water depth (Lorenson et al., 2016). Thus, our methane concentrations fell well within the range reported for other arctic river and coastal systems. A more detailed comparison with temperate and tropical environments is discussed below, in the context of the diffusive methane flux, as most reviews rely on methane emissions rather than on concentrations (Stanley et al., 2016; Ortiz-Llorente and Alvarez-Cobelas, 2012).

Our classified water masses were separated by a strong pycnocline, so different parameters influenced the corresponding methane distributions. In polar water with a median methane concentration of 26 nmol $L^{-1}$, linear regression analysis revealed that the methane concentration of the surface sediments was the only important factor determining the methane concentration in the water above. We assumed that this methane mostly originated from methane flux out of the sediment. In the shallow Chucki Sea, methane was also arising from the decomposition of organic carbon from the seafloor (Fenwick et al., 2017). A further source of methane for bottom waters is submarine groundwater discharge, as shown for two Alaskan sites (Lecher et al., 2016). However, the low tidal amplitude, low topographic relief and low precipitation in the present study area do not favour a high groundwater input to the Lena delta. Highly active methane seeps are also reported for this region (Shakhova et al., 2014), and methane ebullition could be another reason for the observed high methane concentrations. No sonar data were available for our cruise, so we do not have any information on seep activity. In addition, our data do not show an increased methane concentration at the pycnocline, where entrapped gas bubbles could dissolve (Gentz et al., 2013), so ebullition is unlikely to be a significant source of methane. However, we were unable to conduct isotope analysis to verify the origin of the bottom water methane.

At the surface of riverine water, several methane sources are possible, including in situ production, input from bottom water and riverine input. We showed a positive correlation between the methane concentrations in riverine water and temperature and a negative correlation with oxygen concentration. These correlations could be related to the degradation processes that ultimately lead to methanogenesis, as these processes are enhanced by temperature and are oxygen consuming. The removal of dissolved organic carbon occurs primarily at the surface layer, where about 50% of the terrestrial organic material is mineralised (Kaiser et al., 2017). For lakes and oceans, a link is reported between photosynthesis and methane production (Tang et al., 2014), or even evidence of methane production by marine algae (Lenhart et al., 2016), and this activity results in oversaturated methane concentrations in surface waters. Dimethylsulfoniopropionate (DMSP), which is formed as an

osmoprotectant and antioxidant in microalgae, could also be a source of in situ methane production (Florez-Leiva et al., 2013). However, the contributions of photosynthesis and DMSP production to in situ methane concentrations remain to be established.

Methane input from bottom water to surface water will not be important at the deeper stations (e.g. T1-1304 – 07), as the strong water column stratification will limit any exchange processes. However, at the shallower stations (< 8 m, i.e. the coastal stations of the transects), where the water column was mixed, sediments may be the source of the surface water methane.

Another source of methane might be the water of the Lena River itself, as rivers or estuaries are thought to export methane-rich water into coastal seas. Methane concentrations in the Bykowski Channel of the Lena River are, on average, $58 \pm 19$ nmol $L^{-1}$ (Bussmann 2013, and unpublished data from 2012 and 2016). We did find elevated methane concentrations near the coast, but salinity and methane concentrations were not correlated in either the separate water masses or the whole data set (i.e. we observed no dilution of methane-rich river water with methane-poor marine water; Figure 7), in agreement with our previous findings (Bussmann, 2013a). For other estuaries, a complex pattern of increasing/decreasing methane concentrations versus salinity has been presented (Borges and Abril, 2012). However, none of the currently proposed schemes seems applicable to our data. One reason for the lack of significant correlation between salinity and methane concentrations could be the presence of another source of freshwater containing only minor methane amounts. In contrast to other estuaries, arctic estuaries are ice covered for about two thirds of the year, and the seasonal freezing and melting of ice has a strong impact on the water budget. The freezing of sea water results in brine formation with strongly increased salinity, while its melting results in a freshwater input (Eicken et al., 2005). To a lesser extent, this also holds true for freshwater ice. In 1999, the river water fraction in ice-cores near our study area ranged from 57 to 88% (Eicken et al., 2005). Thus, melting of this ice in spring would provide an additional freshwater input. Not much is known about methane concentrations in ice, but a recent study on sea-ice in the East Siberian Sea (Damm et al., 2015) indicated that the methane concentrations are probably lower in this melt water than in the river freshwater. The melting of ice in springtime could therefore add a freshwater input with a minor methane concentration. This additional aspect of the water budget in ice-covered estuaries might explain the missing relationship between salinity and methane concentration in the Lena delta.

### 4.2 Methanotrophic activity and the methanotrophic population

We measured an overall median MOX of 0.32 nmol $L^{-1} d^{-1}$, ranging from 0.03 to 5.7 nmol $L^{-1} d^{-1}$. In other coastal seas, comparable values have been observed, with a median of 0.82 and 0.16 nmol $L^{-1} d^{-1}$ for the coastal and marine parts of the North Sea, respectively (Osudar et al., 2015), 0.1 nmol $L^{-1} d^{-1}$ at the surface of the central North Sea (Mau et al., 2015) and 1 to 11 nmol $L^{-1} d^{-1}$ for Eckernförde Bay in the Baltic Sea (Steinle et al., 2017). In polar waters, off the coast of Svalbard and unaffected by ebullition sites, values of 0.26 to 0.68 nmol $L^{-1} d^{-1}$ (Mau et al., 2017) and $0.5 \pm 1$ nmol $L^{-1} d^{-1}$ (Steinle et al., 2015) have been reported. Thus, our values are well within the reported ranges reported for coastal and polar MOX. However, at the source of the riverine water (i.e. the Lena River itself), much higher MOX (median = 24 nmol $L^{-1} d^{-1}$) have been observed (Osudar et al., 2016). The first order rate constant used for modelling the methane flux in the Laptev Sea ranged from 18116 $d^{-1}$ to 11 $d^{-1}$ (Wahlström and Meier, 2014). Based on our data, we suggest more realistic first order constants (and

turnover times) of 0.01 d$^{-1}$ (91 d) in riverine water, 0.006 d$^{-1}$ (167 d) in mixed water and 0.03 d$^{-1}$ (36 d) in polar water.

In the riverine water, MOX and fractional turnover rates were correlated with temperature (ranging from 7 to 11°C), while the other water masses showed no such correlation. The influence of the methane concentration on MOX was also most pronounced in riverine water ($r^2 = 0.98$). In polar water, methane concentration had a much lower influence ($r^2 = 0.56$).

The described method of qPCR and the use of water column specific primers (Tavormina et al., 2008) gave a
relative abundance of MOB in our study ranging from 0.05 to 0.47% (median 0.16%), which is equivalent to $4 \times 10^4$ to $3 \times 10^6$ cells L$^{-1}$ (median $6.3 \times 10^4$ cells L$^{-1}$). In a marine area with no methane seep, 2 to 90 copies of MOB DNA per mL, equivalent to 1 to $45 \times 10^3$ cells L$^{-1}$, have been reported (Tavormina et al., 2010), assuming two copies of the pmoA gene per cell (Kolb et al., 2003). In the Lena River, the number of MOBs ranges from 1 to $8 \times 10^3$ cells L$^{-1}$ (Osudar et al., 2016). In the boreal North Sea, a broad range of $0.2 \times 10^3$ to $8 \times 10^8$ cells L$^{-1}$
were found (Hackbusch, 2014). These studies all performed qPCR with the same primers used in the present study, and our numbers are within the upper range of the reported values. The use of CARD-FISH seems to give higher numbers of MOBs, at 3 to $30 \times 10^6$ MOB cells L$^{-1}$ for the polar waters off the coast of Svalbard (Steinle et al., 2015) and $1 \times 10^6$ cells L$^{-1}$ for the surface waters at the Coal Oil Point seep field in California (Schmale et al., 2015).

We found no correlation between methane-related parameters (methane concentration, MOX and k') and either cell number or relative abundance of MOBs, but we found correlations with parameters that are important for establishment of a heterotrophic bacterial population, such as DOC, temperature and salinity (Lucas et al., 2016). For this reason, we have to assume that our qPCR assays also detected cells that were not active. This assumption is supported by the finding that even when MOX was not detectable, we still detected MOB-DNA in
our samples. Conversely, when MOB-DNA was not detectable, we were still able to measure MOB activity as MOX. This could be due to the failure of our qPCR protocol to amplify some of the MOBs present in our samples. The primer set used in this study is the most frequently used; however, a few other primer sets are available for amplification of specific monooxygenase genes in several subgroups that are not targeted with the primer set used here (Knief, 2015). Thus, these subgroups – for example, *Verrucomicrobia* or the anaerobic
methanotrophic bacteria of the NC10 phylum, and others (Knief, 2015) – would not be quantified in our study. Similarly, dormant MOBs might be present, whose DNA would be detected even though the cells were not active (Krause et al., 2012). However, we can state that the different water masses had significantly different abundances of MOBs, with the highest abundance in riverine water and the lowest abundance in polar water.

The MISA method used in the present study generated the first successful fingerprinting of the methanotrophic
population in a polar estuary. Until now, only one study has applied MISA to environmental samples, and two OTUs were described in that marine study (Tavormina et al., 2010). The first group, OTU-1, has a broad distribution and belongs to a known group of gammaproteobacteria. In our study, OTU-445, assigned to group OTU-1, was distributed equally in all the different water masses we analysed. The second group, Group-Z, is not as abundant and belongs to a group of MOBs of unknown lineage and function (Tavormina et al., 2010). In the
present study, OTU-535, which was assigned to Group-Z, preferred the non-polar environment, whereas OTU-485, which was assigned to the *Methylococcus* group, showed no specific associations. We conclude that the

methanotrophic populations differ in polar versus river/mixed water: some OTUs were absent from polar water and one OTU had a clear association with polar water. The populations in riverine and mixed water were very similar. One subset of OTUs identified in this study could not be linked to any known MOBs. A useful, if

challenging, future task would therefore be to isolate and describe these as yet unidentified polar MOBs to help in determining MOB diversity. Further insight could be gained by next-generation sequencing, which would provide an in-depth view of population structure. Meta-genomic and meta-transcriptomic analyses could also help to identify functional genes and reveal which MOB types are truly active and which are dormant.

The ecological traits determined in the present study can be summarised as follows. We observed two distinct

methanotrophic populations with different characteristic in the riverine versus polar water masses. In polar water, the methanotrophic activity was influenced by the nitrogen content and very little by the methane concentration. The relative abundance and estimated diversity of MOBs was lower in polar water than in riverine water. Thus, this polar population was well adapted to the cold and methane-poor polar water environment, but it was limited by the nitrogen content. The MOBs in the polar population were lower in relative abundance and

had a lower estimated diversity than the MOBs in the riverine population, but these microorganisms were quite efficient at reaching a MOX comparable to that observed in riverine water. By contrast, the riverine population, despite its higher relative abundance and estimated diversity, showed a methanotrophic activity that was limited by temperature and methane concentrations. Consequently, this population was not very efficient when subjected to sub-optimal temperatures and substrate concentrations.

Methane concentration and nitrogen availability are strong driving forces that shape MOB community composition and activity (Ho et al., 2013). Interactions with other heterotrophic bacteria can further influence the features of the methanotrophic community (Ho et al., 2014). Removal and degradation of dissolved organic matter occurs mainly at the surface and in riverine water (Gonçalves-Araujo et al., 2015), so this may lead to additional enrichment of the methanotrophic population in riverine water. We also assume that the riverine

environment is subject to more environmental changes (salinity, light and temperature) when compared to the polar one. Changes in salinity have different impacts on sensitive and non-sensitive MOBs, thereby shaping the methanotrophic community (Osudar et al., in press 2018). In contrast to our more diverse riverine population, the methanotrophic population in the Lena River proper was characterised by a rather homogenous community (Osudar et al., 2016). However, the classical concept of the r- and k-strategists has today been replaced by the

competitor–stress tolerator–ruderal functional classification framework (Ho et al., 2013). Thus, the type Ia MOBs found in the present study, which respond rapidly to substrate availability and are the predominant active community in many environments, can also be classified as competitors (C) and competitor–ruderals (C–Rs) (Ho et al., 2013).

### 4.3    Diffusive methane flux

The calculation of the diffusive methane flux requires several parameters, including the atmospheric methane concentrations. According to the database of the meteorological station in Tiksi, these ranged from 1.896 to 1.911 ppm and are within the range of values previously reported (1.879 ppm) in the summer of 2014 for the outer ice-free Laptev Sea (Thornton et al., 2016). By contrast, our wind speed was somewhat higher (4.2 ± 2.2

490 m/s) than the 2.9 ± 1.9 m/s reported previously (Thornton et al., 2016). This difference would result in slightly higher equilibrium concentrations and a higher gas exchange coefficient in our study.

The gas exchange coefficient, k, is a more critical value and is also more difficult to assess. No current method is totally satisfactory for quantifying k in estuaries, and its calculation remains a matter of debate (Borges and Abril, 2012). In their review, Borges & Abril (2012) report an approximate range for $k_{600}$ of < 10 up to 30 cm/h

(< 2.4–7.2 m/d). For the North Sea in winter, much higher values were obtained (7–62 cm/h = 17–150 m/d) (Nightingale et al., 2000). Similar values were reported for a bay in the Baltic Sea, at around 7 cm/h = 17 m/d (Silvennoinen et al., 2008), but lower values were reported for a Japanese estuary in summer (0.69–3.2 cm/h = 1.7 -7.7 m/d) (Tokoro et al., 2007). Our values for $k_{600}$ ranged from 0.37 to 3.17 m d⁻¹, with a median of 1.05 m d⁻¹. Thus, our $k_{600}$ values fell within the lower range reported in the literature.

Considering all the assumptions and additional data, we calculated a median diffusive methane flux of 24 $\mu$mol m² d⁻¹, ranging from 4 to 163 $\mu$mol m² d⁻¹. Our data lay well within the range of data reported from previous studies within this area (Table 5; Bussmann, 2013a; Shakhova and Semiletov, 2007). Wahlström and Meier (2014) applied a modelling approach that resulted in even lower methane fluxes (Table 5).

The area off the Svalbard coast is another polar region with an appreciable scientific focus. A comprehensive

study by Myhre et al. (2016) calculated a median methane flux of only 3 $\mu$mol m² d⁻¹, which is supported by a median methane flux of 2 $\mu$mol m² d⁻¹ for the coastal waters of Svalbard (Mau et al., 2017), and this value lies within the previously reported range of 4 to 20 $\mu$mol m² d⁻¹ (Graves et al., 2015; Table 5). For the North American Arctic Ocean and its shelf seas, rather low methane fluxes of 1.3 $\mu$mol m² d⁻¹ have been reported (Fenwick et al. 2017).

Our two stations with the highest methane fluxes had flux values similar to those reported for the North Sea with a mixed water column. In the North Sea, the stratification of the water column in the summer significantly reduces the diffusive methane flux, even at an active seep location (Mau et al., 2015). The values for a stratified fjord in the Baltic Sea are comparable to those of the North Sea (Steinle et al. 2017). However, in the southern North Sea, which has a mixed water column, very high methane fluxes (> 200 $\mu$mol m² d⁻¹) are reported, which

are mainly related to organic-rich sediments (Borges et al., 2016). A summary study of European estuaries reported an average methane emission of 118 $\mu$mol m² d⁻¹ (Upstill-Goddard and Barnes, 2016).

Table 5 shows a comparison of our methane emission rates with those reported from other polar sites, as well as some temperate ones. Methane emissions in polar sites seem somewhat lower than those found in temperate sites; however, even within the polar environments, a broad range of emission occurs. A worldwide comparison

of riverine and aquatic methane emissions, presented by Stanley et al. (2016) and Ortiz-Llorente & Alvarez-Cobelas (2012), revealed no correlation between methane emissions and latitude. This finding contrasts with the review by Borges and Abril (2012) comparing worldwide estuaries, where an increase in methane emissions was evident from estuaries at high latitudes, as well as from tidal systems. (Notably, the Lena delta matches both of these classifications.) No overall pattern of controlling factors of methane emission were revealed by Ortiz-

Llorente and Alvarez-Cobelas (2012); thus, the authors concluded that local studies are vital for assessing methane emission and its controlling factors.

The presence and strength of a pycnocline is especially critical in the control of methane emission, as this emission is much stronger from environments without stratification (Borges et al., 2017) than from stratified

systems where MOX can consume part of the methane (Mau et al., 2015). Temperature is another important environmental control factor, as methane production is very temperature sensitive (i.e. methanogenesis is higher at higher temperatures). Consequently, tropical and temperate regions would be expected to show higher methane concentrations and emissions (Borges et al., 2017; Lofton et al., 2014) while polar regions would have lower concentrations and emissions. However, as methane oxidation is only somewhat influenced by temperature, this may offset methane consumption versus methane production in polar areas (Lofton et al., 2014), thereby resulting in lower methane concentrations overall in polar regions. Thawing permafrost is another potential contributor to the polar methane cycle, although this remains a controversial issue (Overduin et al., 2015; Shakhova et al., 2010). A previous molecular approach identified salinity, temperature and pH as the most important environmental drivers of methanogenic community composition on a global scale (Wen et al., 2017). However, the mechanisms by which changes in these factors influence the methanogenesis rate remain elusive, due to the lack of studies that combine methane production rates with community analyses (Wen et al., 2017).

In contrast to these bottom-up calculations, very few studies have focused on the atmospheric methane concentrations in the study area (Thornton et al., 2016; Shakhova et al., 2014; Shakhova et al., 2010) or in polar regions in general (Myhre et al., 2016). The top-down calculations of methane flux seem to be higher than the bottom-up calculation, at 94 and 200–300 $\mu$mol m$^2$ d$^{-1}$, respectively (Thornton et al., 2016; Myhre et al., 2016). Ebullition of methane from the sediment in this area is also reported, resulting in methane fluxes that are 1–2 orders of magnitude higher than the calculated values (Table 5). Previous examinations of methane released by ebullition did not find any isotopic evidence of oxidation; thus, this methane will almost exclusively be released into the atmosphere (Sapart et al. 2017). However, whether this ebullition really results in elevated atmospheric methane concentrations remains a matter of debate, as this fingerprint has not been detected by others (Thornton et al., 2016; Berchet et al., 2015). Overall, methane emissions from the East Siberian Arctic shelf seem relatively insignificant when compared to methane emissions from wetland and anthropogenic sources in eastern Siberia (Berchet et al., 2015).

### 4.4    Role of microbial methane oxidation versus diffusive methane flux

We estimated the role of methane oxidation and diffusive methane flux for the methane inventory in the Lena delta by calculating the total methane inventory (see Sect. 2.7), as well as the total methane oxidation and total diffusive flux of this area. When the total methane inventory was set to 100%, a median of 1% (range 0.3–3.8%) was consumed within one day by bacteria within the system, while a median of 8% (1–47%) left the system and entered the atmosphere. A similar estimation has been made for the coastal waters of Svalbard (Mau et al., 2017), where a much higher fraction of the dissolved methane (0.02–7.7%) was oxidised, and only a minor fraction (0.07%) was transferred into the atmosphere. However, the water in this region was much deeper; thus, the ratio of water volume (including the methane oxidation activity) to the surface area (including the diffusive methane flux) was much larger. Another polar study conducted off the coast of Svalbard suggested that about 60% of the methane in the bottom water is oxidised before it can mix with intermediate or surface water (Graves et al., 2015). For the coastal waters of the Baltic Sea, water column stratification is also crucial, much more methane is oxidized in a stratified water column compared to a mixed water column (Steinle et al. 2017).

Our estimate of methane flux is a static one and does not take into account the currents and spreading of the freshwater plume. In estuaries, the residence time of the water (as influenced by water discharge and tidal force) also influences the efficiency of the estuarine filter (Bauer et al., 2013). The bulk of the freshwater from the Lena River stays in the eastern Laptev during the summer season (Fofonova et al., 2015). However, changing atmospheric conditions render the freshwater content in the Laptev sea shelf highly time-dependent and turbulent (Heim et al., 2014). The simulations performed by Wahlström and Meier (2014) revealed the importance of the methane oxidation rate constant and the crucial necessity of obtaining an in situ measurement of it. The concentration of methane in the river runoff and the methane flux from the sediment are also statistically significant and important factors for determining the sea to air flux of methane (Wahlström and Meier, 2014).

## 5 Conclusions

In the context of the predicted and ongoing warming of the Arctic regions, two main factors are expected to change for coastal arctic seas. One is the hydrographic regime, which will experience a greater freshwater input and stronger stratification (Bring et al., 2016). The second is thawing of the permafrost, which will increase the fluxes of carbon and nutrients into the coastal arctic region. The released material can then be dissipated by several routes: it can be degraded directly into greenhouse gases, it can fuel marine primary production, it can be buried in nearshore sediments or it can be transported offshore (Fritz et al., 2017).

Based on our data, we postulate the following changes in the methane cycle in the Lena delta. An increased freshwater input does not necessarily lead to higher methane concentrations in the study area, as we found no evidence of a direct methane input by the Lena River. Instead, a more complex pattern of methane input develops. An increased freshwater input would also result in more nutrients and increased turbidity of the water. The former would stimulate primary production, while the latter would reduce it. Thus, if the altered primary production would lead to an increased degradation of organic material and subsequent methanogenesis or to an altered in situ methane production in surface riverine water is not clear yet. However, the methanotrophic population in this water mass is very diverse and is expected to adjust to a changing environment and respond well to increasing water temperatures.

A strong stratification in polar water, together with increased inputs of particulate organic material to the bottom water, probably increases the degradation processes, as well as the methane concentrations in the surface sediment and the water column above it. The polar methanotrophic population in our study was quite efficient and we predict that it can compensate for any increase in methane concentrations. However, increases in storm frequency or strength will disrupt the stratification of the water column and promote mixing of the different water masses. In our study, we showed that the conditions in a mixed water mass were unfavourable for MOBs and resulted in an approximately 4-fold reduction in the MOX. An increase in methane emissions after a storm has already been reported in this study area (Shakhova et al., 2014).

The methane sinks in the present-day water column of the Lena delta are rather weak. Consequently, 1% of the methane inventory is oxidised per day and 8% diffuses into the atmosphere. The Lena delta water masses therefore represent a strong methane source for the waters of the Laptev Sea and the central Arctic Ocean, whereas they make only a limited contribution to atmospheric methane levels.

**Acknowledgements**

The authors acknowledge the Captain and the crew of the RV *Dalnie Zelentsy*. We are thankful to the logistics department of the Alfred Wegener Institute, particularly W. Schneider. Special thanks go to N. Kasatkina and D. Moiseev from the Murmansk Marine Biological Institute for offering laboratory support. We also want to thank

Ellen Damm for fruitful discussions and Patricia Tavormina for help in setting up the MISA. The methane-related data set is available at www.pangaea.de, doi:10.1594/PANGAEA.868494, 2016.

**Figures and Tables**

**Figure 1. Map of the study area in September 2013 and sampling locations, with four transects heading from near shore to offshore. The dashed lines delineate the area used for the budget calculation.**

**Figure 2. Salinity (A) and methane (B, in nmol L$^{-1}$) distributions versus depth and distance from the shore for Transect 1. In (A) the water masses are also indicated, defined as riverine water (salinity (S) < 5), mixed water (5 < S >20) and polar water (S > 20). The grey bars indicate the location of the stations. In (B), for stations with very high methane concentrations, the values are annotated in the figure.**

**Figure 3. Methane concentrations in nmol L$^{-1}$ at the surface of the study area. For stations with very high methane concentrations, the values are annotated in the figure.**

**Figure 4. Correlation between the methane concentration in bottom water and the concentration in the**

**underlying sediment for all stations (r$^2$ = 0.62, p < 0.001, n= 33). Two very high values from station TIII-1304 were excluded from the analysis.**

**Figure 5. Logarithm of the methane oxidation rates in nmol L$^{-1}$ d$^{-1}$ in surface (A) and bottom (B) water around the Lena delta.**

**Figure 6. Relative abundance of methanotrophic DNA (as %MOB-DNA) in surface (A) and bottom (B) water around the Lena delta. For stations with very high methane concentrations, the values are annotated in the figure.**

**Figure 7. Methane concentration versus salinity for riverine water (open circles), mixed water (diamonds) and polar water (open squares). The dotted line indicates a regression line for all data points (r$^2$ = 0.01, p = 0.7, n = 99).**

**Appendix Figure A1. Salinity in surface waters around the Lena delta.**

.

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

Table 1. The median values of important parameters (methane concentration and oxidation rate, fractional turnover rate k', turnover time, relative abundance and diversity of methanotrophs) in the different water masses. A one-way analysis of variance (ANOVA) was performed on the log-transformed data to test for significant differences between the water masses.

| | Median for Riverine water | Median for Mixed water | Median for Polar water | DF / p[1] |
|---|---|---|---|---|
| $CH_4$ [nmol $L^{-1}$] | 22 | 19 | 26 | **94 / 0.03 *** |
| MOX [nmol $L^{-1}$ $d^{-1}$] | 0.419 | 0.089 | 0.400 | 68 / 0.18 |
| k' [d] | 0.011 | 0.006 | 0.028 | **68 / < 0.001 *** |
| Turnover time (d) | 91 | 167 | 36 | |
| %MOB | 0.81 | 0.19 | 0.03 | **23 / <0.001 *** |
| estimated diversity [OTUs / station] [2] | 4 | 3 | 2 | **23 / 0.01 **** |

1 results of the ANOVA with degrees of freedom (DF) and level of significance (p).

2 operational taxonomic unit (OTU)

Table 2. Linear correlation between the methane concentration versus different environmental parameters seperated for the three water masses. Analysis was performed with log transformed data; the $r^2$-values, the level of significance (p) and the positive or negative correlation (+/-) are shown. Bold numbers indicate a significant

correlation (p<0.05).

| | Riverine water (n = 13) | Mixed water (n = 22) | Polar water (n = 24) |
|---|---|---|---|
| Temperature | **(+) 0.38 / 0.02** | (+) 0.003 / 0.74 | (-) 0.10 / 0.04 |
| Salinity | (-) 0.23 / 0.13 | (+) 0.03 / 0.25 | (-) 0.0001 / 0.93 |
| $O_2$ | **(-) 0.73 / <0.001** | (-) 0.02 / 0.36 | (-) 0.006 / 0.65 |
| $DOC^1$ | (+) 0.002 / 0.89 | (+) 0.01 / 0.31 | (-) 0.0003 / 0.94 |
| $TDN^2$ | (-) 0.0006 / 0.95 | **(+) 0.27 / 0.01** | (+) 0.11 / 0.12 |
| Sediment $CH_4$ | n.d. | n.d. | **(+) 0.33 / < 0.001** |

n.d. not determined due to insufficient number of data points

1 dissolved organic carbon (DOC)

2 total dissolved nitrogen (TDN)

Table 3. Linear correlation between the methane oxidation rate (MOX) and the fractional turnover rate (k')
versus different environmental parameters seperated for the three water masses. Analysis was performed with
log transformed data; the $r^2$-values, the level of significance (p), and the positive or negative correlation (+/-) are
shown. Bold numbers indicate a significant correlation (p<0.05).

| | Riverine water (n = 6) | | Mixed water (n = 9) | | Polar water (n = 11) | |
|---|---|---|---|---|---|---|
| | MOX | k' | MOX | k' | MOX | k' |
| Temperature | **(+) 0.77 / 0.02** | **(+) 0.84 / 0.01** | (+) 0.01 / 0.77 | (+) 0.004 / 0.87 | (-) 0.02 / 0.69 | (-) 0.07 / 0.41 |
| Salinity | (-) 0.30 / 0.26 | (-) 0.43 / 0.16 | (+) 0.30 / 0.12 | **(+) 0.46 / 0.04** | (+) 0.05 / 0.52 | (+) 0.17 / 0.21 |
| $O_2$ | (-) 0.33 / 0.23 | (-) 0.30 / 0.26 | (-) 0.006 / 0.83 | (-) 0.07 / 0.48 | (-) 0.03 / 0.67 | (-) 0.001 / 0.92 |
| DOC[1] | (+) 0.29 / 0.27 | (+) 0.46 / 0.14 | (-) 0.009 / 0.80 | (+) 0.02 / 0.75 | (+) 0.004 / 0.85 | (+) 0.007 / 0.80 |
| TDN[2] | (-) 0.02 / 0.80 | (-) 0.002 / 0.93 | (+) 0.30 / 0.13 | (+) 0.27 / 0.08 | (+) 0.31 / 0.08 | (+) 0.12 / 0.16 |
| Methane | **(+) 0.98 / <0.001** | **(+) 0.96 / <0.001** | **(+) 0.80 / < 0.001** | **(+) 0.73 / <0.001** | **(+) 0.56 / 0.01** | (+) 0.13 / 0.31 |

1 dissolved organic carbon (DOC)

2 total dissolved nitrogen (TDN)

Table 4. The occurrence and association of the MISA OTUs to different water masses, their assignation to known methanotrophic groups and the results of a Kruskal Wallis test for significant differences in occurrence ($*$, $p < 0.05$).

| MISA OTU | Assignation | Riverine | Mixed | Polar | Kruskal Wallis | Association |
|---|---|---|---|---|---|---|
| OTU-557 | | 3 | 3 | **9** | 0.06 | Polar |
| OTU-535 | Group Z ** | **6** | **6** | 3 | **0.02 *** | River /mixed |
| OTU-485 | *Methylococcus capsulatus ***  | 3 | 2 | 2 | 0.4 | |
| OTU-460 | | **3** | **3** | 0 | 0.06 | River /mixed |
| OTU-445 | OPU-1 ** | 4 | 3 | 4 | 0.5 | |
| OTU-398 | | **1** | 0 | 0 | 0.2 | River |
| OTU-362 | | **4** | **5** | 2 | 0.1 | River /mixed |
| Median number of OTUs / sample | | 6 | 5 | 4 | **0.02*** | |

** assignation according to Tavormina et al. (2010)

*** assignation according to Schaal (2016)

Table 5. Comparison of diffusive methane flux from the water column into the atmosphere of this region and temperate and polar shelf seas (in $\mu$mol m$^2$ d$^{-1}$).

| Authors | Area | Range | Median |
|---|---|---|---|
| Calculated from dissolved methane concentrations (bottom-up) | | | |
| This study | Lena delta<br><br>(2 coastal stations of Transect 4) | 4–163 | 24<br><br>536 |
| Bussmann, 2013a | Buor-Khaya Bay | 2 -85 | 34 |
| Shakhova and Semiletov, 2007 | Northern parts of Buor-Khaya Bay | 4–8 | |
| Wahlström and Meier, 2014 | Modelled flux for Laptev Sea | 6 ± 1 | |
| Mau et al., 2015 | North Sea with stratified water column in summer | 2 -35 | 9 |
| Mau et al., 2015 | North Sea in winter, including methane seepage | 52–544 | 104 |
| Borges et al., 2016 | Southern North Sea, summer 2010, near shore | 426 ± 231 | |
| Steinle et al., 2017 | Eckernförde Bay, Baltic Sea | 6–15 | 8 |
| Myhre et al., 2016 | West off Svalbard with $CH_4$ seepage. | Up to 69 | 3 |
| Mau et al., 2017 | Coastal waters of Svalbard | -17–173 | 2 |
| Graves et al., 2015 | Coastal waters of Svalbard | 4–20 | |
| Fenwick et al., 2017 | North American Arctic Ocean | -0.4–4.9 | 1.3 |
| Calculated, modelled from atmospheric data (top-down) | | | |
| Thornton et al., 2016 | ice free Laptev Sea | | 94 |
| Myhre et al., 2016 | West off Svalbard with $CH_4$ seepage | 207–328 | |
| | | | |
| Shakhova et al., 2014 | Ebullitive flux around Lena delta | 6250 - 39375 | |

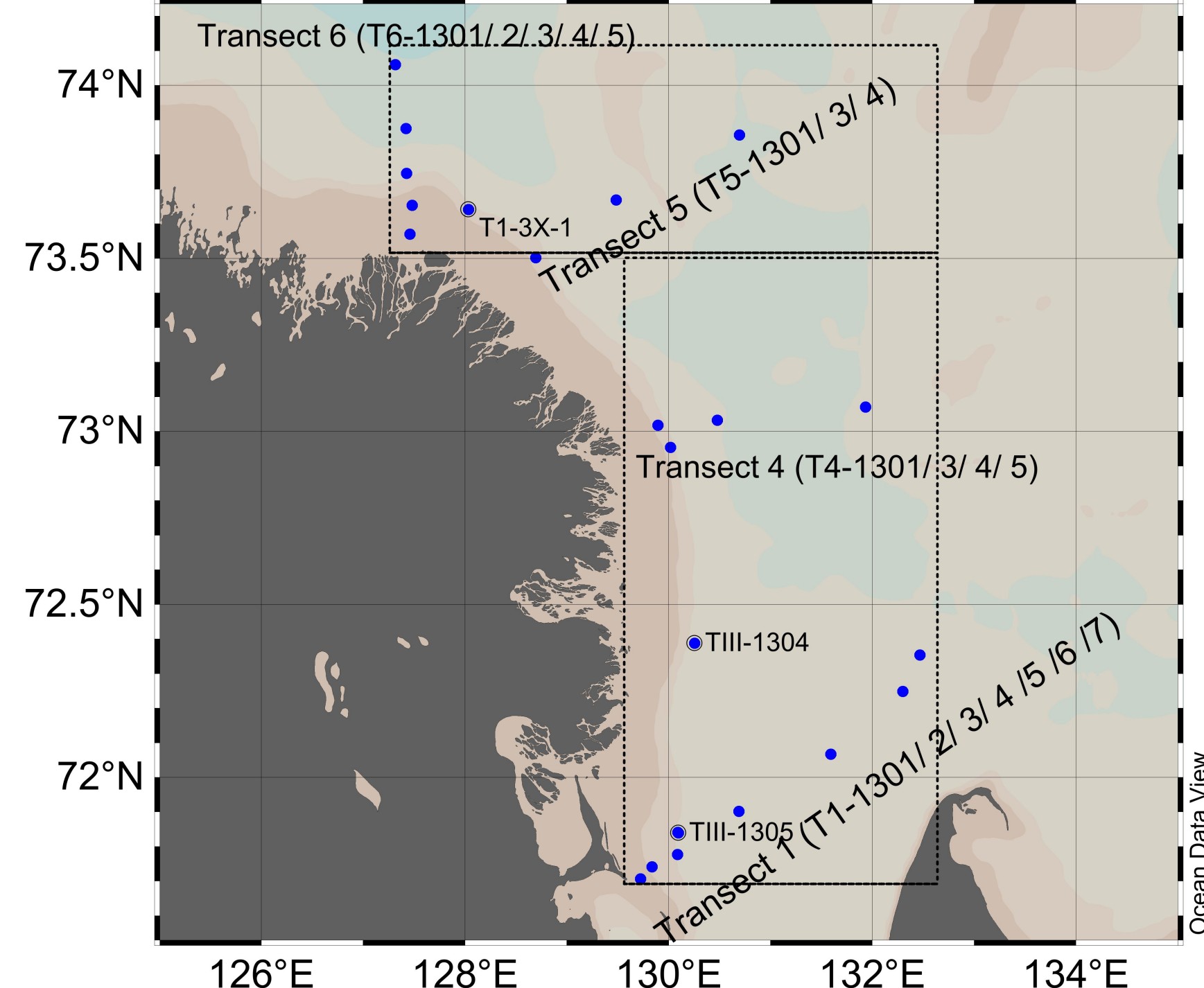

Fig. 1

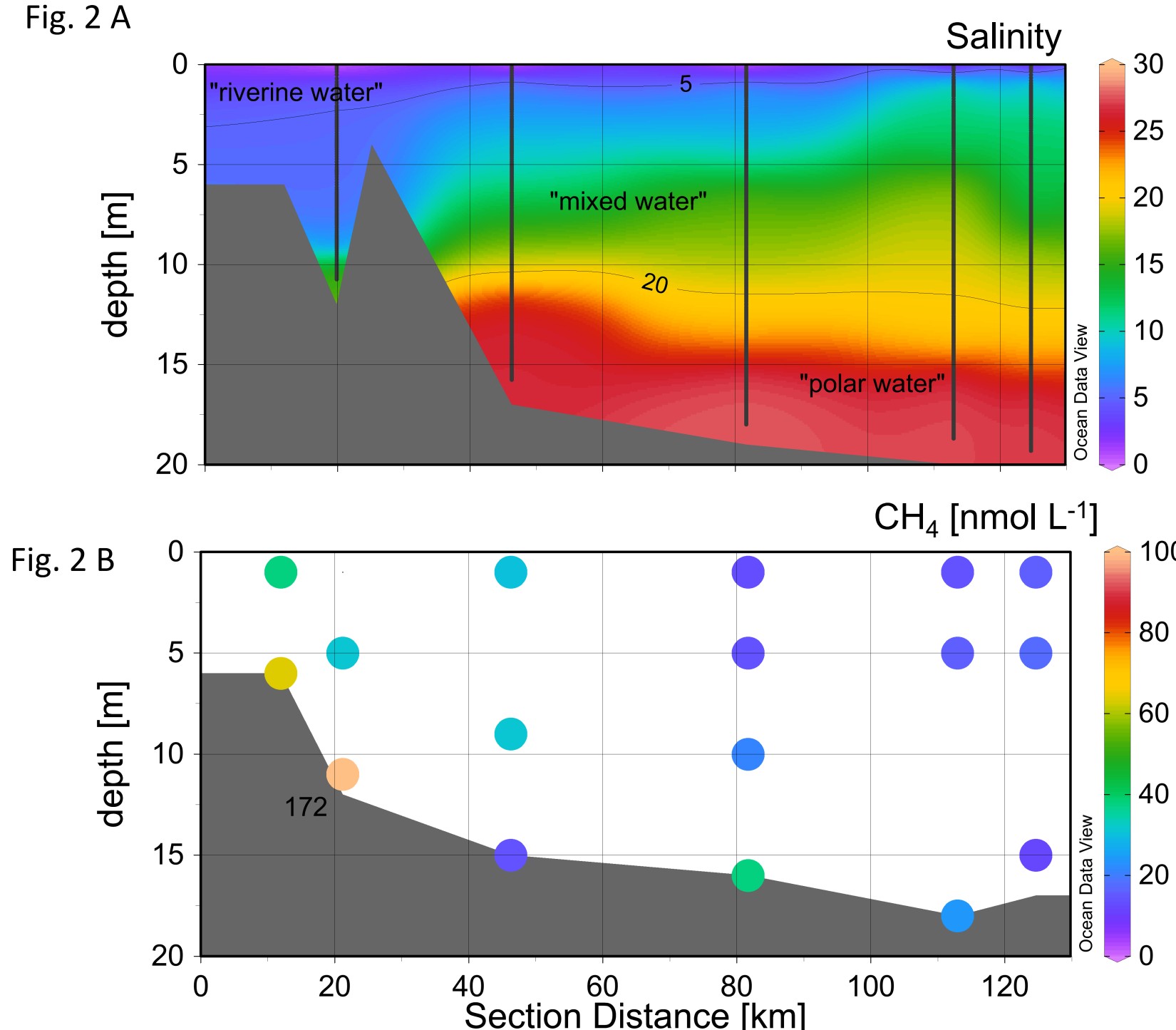

Fig. 2 A

Salinity

"riverine water"

"mixed water"

"polar water"

Ocean Data View

Fig. 2 B

$CH_4$ [nmol $L^{-1}$]

172

Section Distance [km]

Ocean Data View

Fig. 3

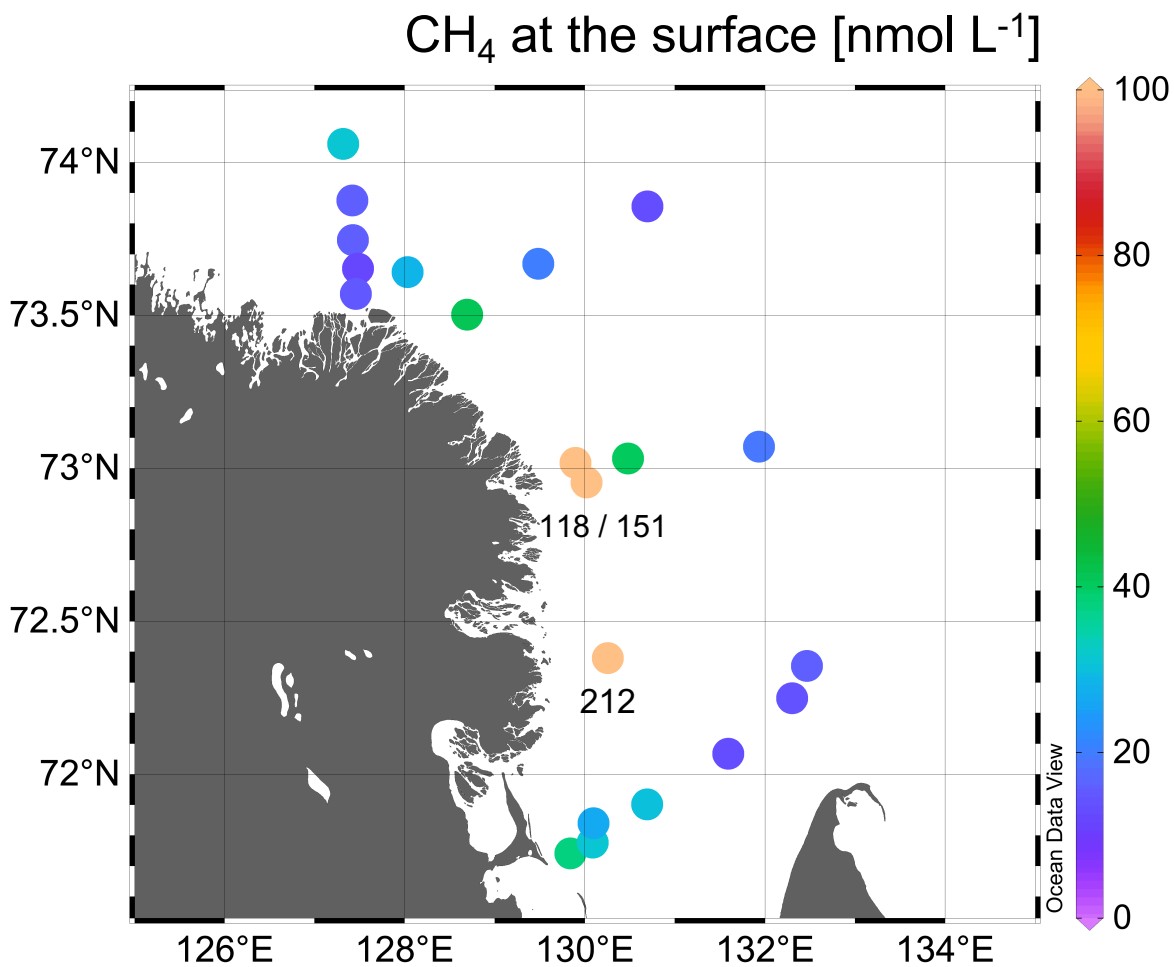

CH$_4$ at the surface [nmol L$^{-1}$]

Fig. 4

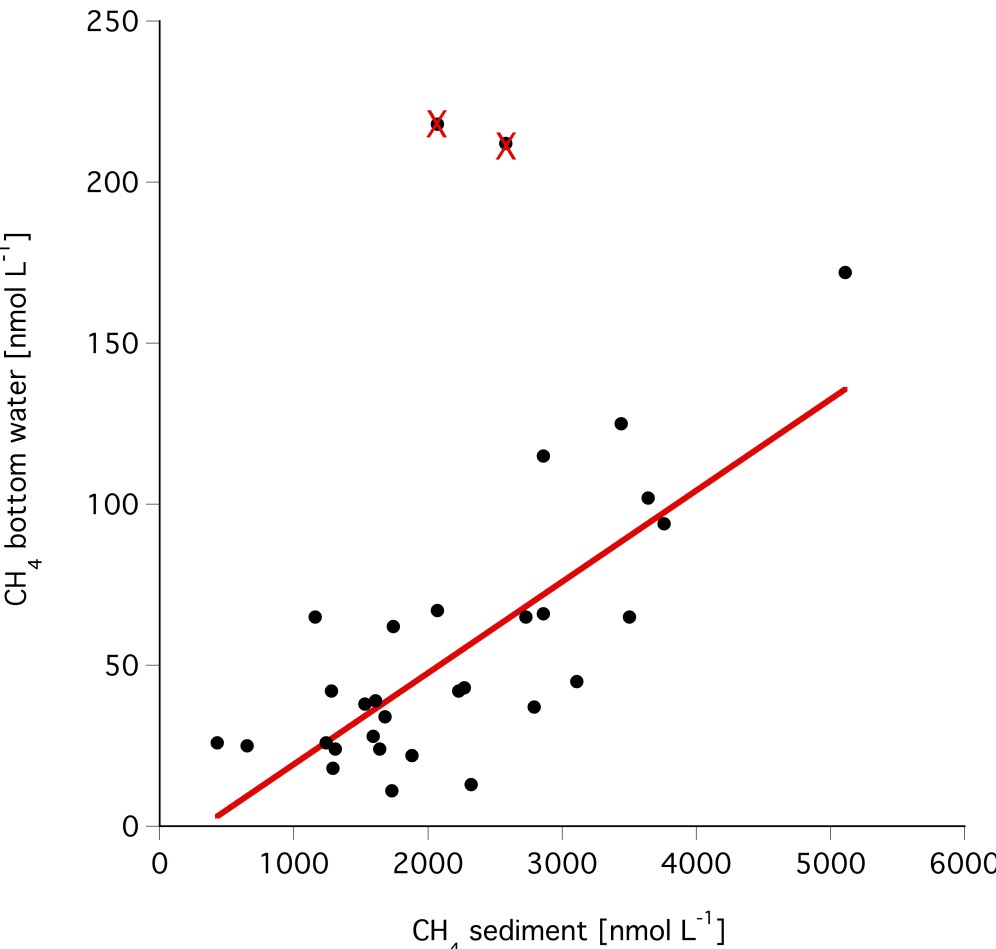

Fig. 5

log MOX at the surface and bottom [nmol L⁻¹ d⁻¹]

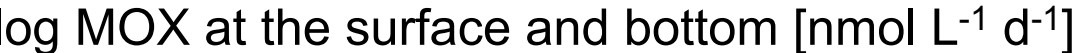

## Fig. 6

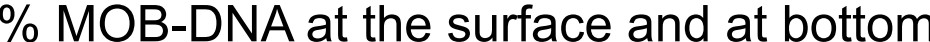

% MOB-DNA at the surface and at bottom

Fig. 7

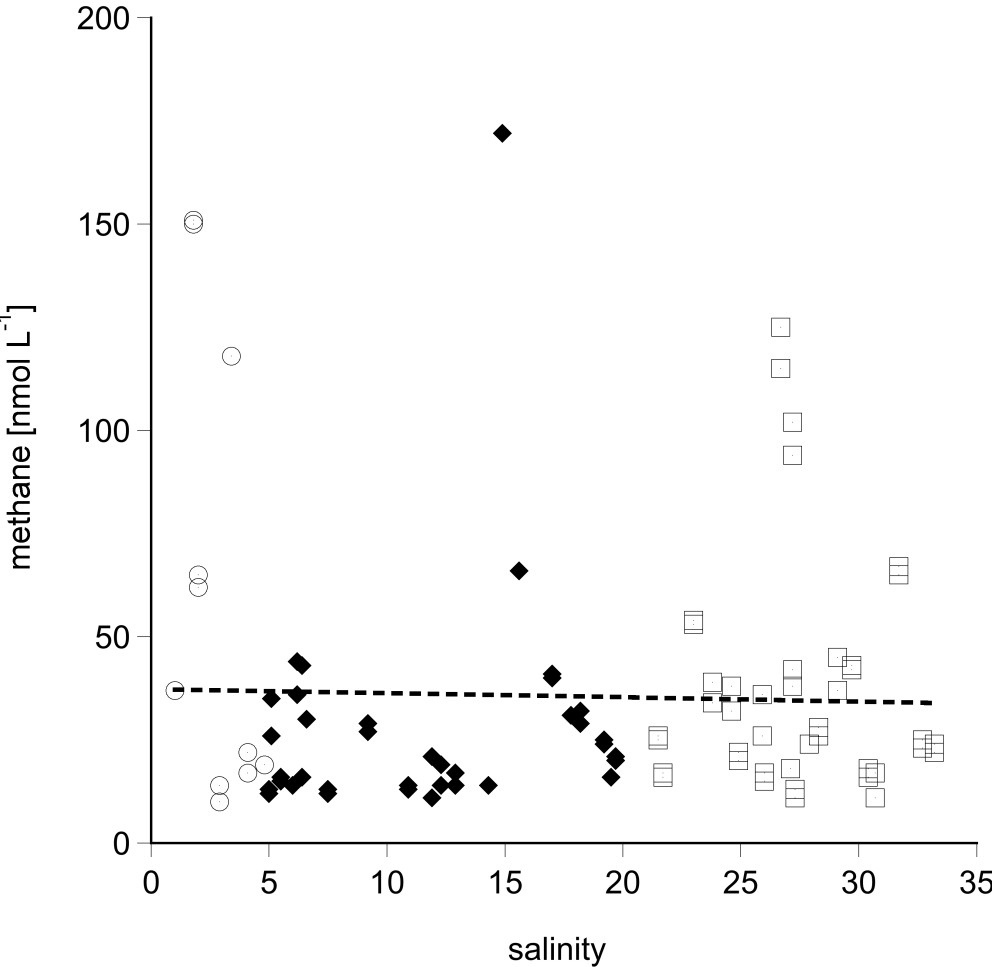

Fig. S1

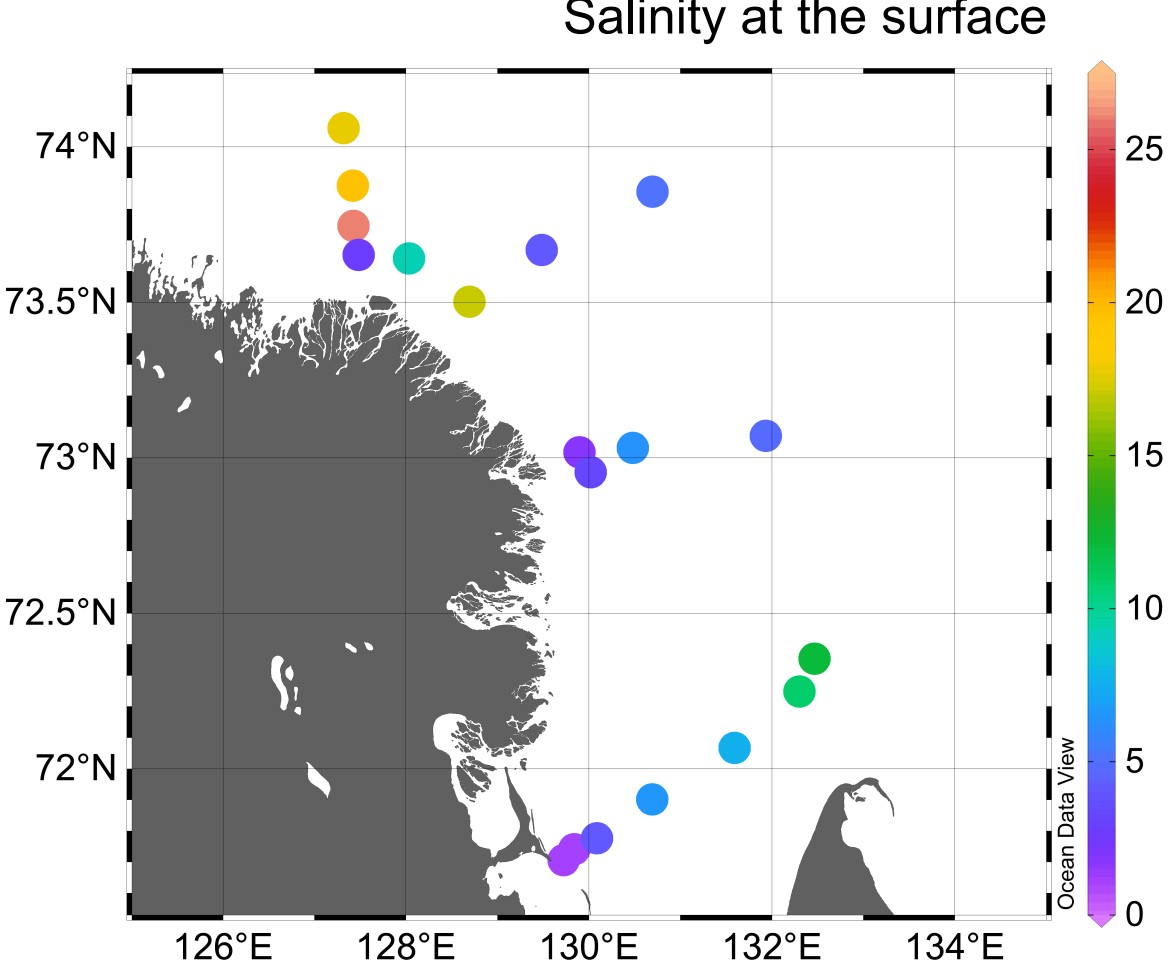