# Peer review of "Methane distribution and oxidation around the Lena Delta in summer 2013"

_Biogeosciences, 2017_

## Referee Comment (RC1) · Anonymous Referee #1 · 20 Mar 2017

In this study, Bussmann et al. examine the distribution and microbial oxidation of methane in the estuarine and coastal waters of the Lena River Delta and Laptev Sea. Here, the authors quantified methane concentrations in different water masses and tested for significant associations with environmental parameters, such as temperature, salinity, DOC, and TDN. Further, they examined methanotrophic populations of bacteria by applying quantitative PCR and intergenic spacer analysis of the particulate methane monooxygenase gene. The authors have developed a hypothesis that separate groups of methanotrophs are differentially specialized in riverine versus polar waters, perhaps in response to temperature, or concentrations of methane and nitrogen. An important finding from this study of a shallow coastal environment is that methanotrophs in this polar environment may consume a small percentage of dissolved methane in the water column.

This study highlights the minimal consumption of methane as a fraction of the dissolved gas, which is in flux to the atmosphere. Moreover, the authors demonstrated that in riverine, mixed, and polar water masses, MOX is significantly tied to methane concentration. The focus here is on the diffusive flux to the atmosphere, but we have no sense of how this diffusive flux compares with ebullition of methane from seeps in the study region. Since this study examined shallow water masses, discussion of any active seep/vent locations in the study area would be helpful, as ebullition is likely to play a major role in methane flux to the atmosphere, and, in turn what fraction of total methane release is available for consumption by MOB.

I am generally supportive of the publication of this study, although mention of marginally significant statistical findings or insignificant results and speculation leading from these should be addressed. In a few cases, grammatical errors and vague language should be rephrased, but addressing these items shouldn't be difficult.

Specific Comments:

L15 – here "methane distribution" refers in parentheses to "headspace", but this isn't a method and it is unclear what is meant. Suggest rewording.

L44 – should read "The source(s) of methane..."

L55 – suggest rewording "water column MOX" to be consistent with first reference to an abbreviation (i.e. "water column methane oxidation (MOX)").

L59 – this sentence seems vague and perhaps unnecessary. Suggest beginning with the following sentence and changing "for some authors" to "In certain studies"

L120 & L132 – change to methane [mono]oxygenase

L133 – were the same primers used here as above?

L224-225 – "This was most pronounced..." the sentence is oddly phrased; suggest rewording.

L230 – 236 "significant" should have a p-value given

L286 – remove mention of OTU "preference" for different water masses, especially where you didn't find a significant trend. Perhaps use phrasing "association" or "link" instead of "preference" throughout.

L379-381 Perhaps MOB with divergent pmoA sequences were not detected with these specific primers? This possibility isn't discussed, but instead speculation was raised that MOB might exist that lack pmoA genes.

L395-396 The statement that "OTUs identified in this study cannot be related to known MOBs" appears to contradict the taxonomic affiliations offered on Line 288. Do you mean that a subset of the OTUs identified in this study cannot be linked to known MOBs?

L415-416 This part is a reiteration of the results on L295. What is the importance of measuring a higher windspeed in comparison to Thornton et al.?

L443 Define (spell out) ESAS; not mentioned elsewhere.

Figure 3. I recommend changing the color for highest methane concentration from pale orange to something that isn't already on your color scale for lower concentrations (e.g., grey or black)

Figure 5. The omission of two data points is mentioned in the main text, but this should also be clearly stated within the figure caption.
* * *

---

## Referee Comment (RC2) · Anonymous Referee #2 · 4 Apr 2017

**Review BGD *Methane distribution and oxidation around the Lena Delta in summer 2013* by Bussmann et al.**

Bussmann et al. present data from a measurement campaign in September 2013 in the coastal area close to the Lena river delta where river water and polar water mix. The activity (qPCR) and the abundance of methanotrophic bacteria was investigated and statistically compared to methane concentrations and physico-chemical parameters in order to determine environmental controls of MOX. Three water masses (river, mixed and polar) were defined previous to statistical analyses. This manuscript employs primers developed by Tavormina et al., which were even improved since the last publications by these authors. The use of these primers to investigate the methanotrophic marine community is quite new and I think that this is the strongest point of this manuscript. Conventional primers often don't cover the marine diversity. I enjoyed reading the manuscript since it is clearly written and everything is well-explained and a wide-range of literature is being put in context with the results of the presented study. There are, however, quite a few formatting/language mistakes. More importantly, I'm missing a more conclusive discussion (see below). If the remarks below can be addressed, most importantly the discussion, this manuscripts presents a solid addition to the current scientific pool of MOX studies and is suitable for publication in BG.

**General remarks:**
1) Did you try to analyze the data statistically without grouping it into different water masses? What are the results then? Or maybe set the salinity borders differently?

2) It would be interesting to do qPCR with sediments samples from the river and coastal area. Especially for the 'outlier station' where authors hypothesize that part of the community got resuspended due to stormy weather. Was this done?

3) The discussion is quite descriptive. I'm missing a more in-depth analysis of the results. For example, the third paragraph of 4.2 is very descriptive. What are the possible reasons that these communities are limited by different factors? Why is the riverine community more diverse? Due to stability? My opinion is that for the MS to be published in BG a less descriptive Discussion part is crucial.

4) A wide range of statistical data is presented. It would be better to discuss the most important findings to avoid confusion of the reader.

**Several small remarks, also with regard to formatting/language mistakes:**
-please check upper/lower case of chemical formulas/mathematical formulas
-abstract line 11: biological "way" sounds a bit strange. Maybe biological sink?
-abstract line 21: riverine, not rivine
-abstract, line 22: "..riverine water TO (not AND).."
-abstract line 17: "..a median OF 28 nM.."
-line 44: hydrate not hydrated
-several times you write 'according to/XX to (XX et al, 1998)'. Please put the parentheses at the right place.

-2.2 why are you using different chemicals (H2SO4 and NaOH) to kill samples for methane analyses for sediment and water samples.

-if you're sampling sediments with a grab sampler for methane analyses, is there not a lot of methane lost on the way up to the ship?

-line 199: remove the 'than'

-line 238: herEby

-if you're correlating MOX to CH4: how can you be sure that's possible since MOX=CH4*k. Isn't what you're calculating then just assessing if k is much smaller than the CH4 concentration (which it generally is).

-line 311: "..seemed to be.."? or there was none?

-line 324: degradation processes? You mean methanogenesis in the sediments?

-line 334 and after: I can't really follow your explanations. Could you rephrase/shorten/write it clearer. I might have missed something but I did not get your point.

-4.2: there was recently a paper published in BG about MOX in coastal environments (Baltic Sea, Eckernförde Bay). Would be good to include it.

-line 356: "..fractional turnover rateS.."

-line 375: "..but more..": what do you mean? More than no correlation? Please rewrite.

-line 380: what's the different from dormant MOB to not active MOB? Do you mean dormant, for instance as endospores? Please write more clearly. Like this, it reads like a repetition from line 376.

-line 403 and 407: limited or influenced? I would prefer a clearer way of writing this.

-line 433: where was Graves et al., 2015 measuring fluxes?

-line 437: did Sapart et al. not measure atmospheric fluxes? Graves et al., 2015 also measured atmospheric methane.

-line 439: remove the ":"

-line 443: what is ESAS?

-line 447: change than to then (also at other places in the MS, please double-check)

-line 451: there was recently a paper published in BG about MOX in coastal environments (Baltic Sea, Eckernförde Bay). Might be interesting to compare the two.

-Figures made with Ocean Data View: Make sampling spots more visible! It would be better not to use the mode where two data-points merge together (interpolation) since there are so few data points.

-Figures: check lower/uppercase

-Table 5: there is not a very good coverage for shelf seas (eg North-Am. Coast, Baltic Sea)! I enjoy this table and it would be good to extend it a bit.

---

## Referee Comment (RC3) · Anonymous Referee #3 · 26 Apr 2017

GENERAL COMMENTS

Bussmann and colleagues report a valuable data-set of dissolved CH4 concentration in the Lena Delta.

It could be useful if authors compare in much more detail their new data-set with older data-sets obtained in the area (Bussmann et al. 2013). As it stands it's unclear what's the added value and novelty of the present ms compared to what was previously published by the authors on the same topic.

The CH4 concentrations in the study area are extremely low compared to other estuarine environments (at lower latitudes), and the spatial gradients are also extremely low given the large salinity gradients. This fundamental difference contains some potentially important information on the functioning of estuaries in high latitudes and deserves to be discussed in light of published CH4 data in other estuaries. Is this due to a low CH4 concentrations in the Lena inner river itself? Any data on the CH4 concentration in the river itself ? If so does it differ from other rivers worldwide (e.g. Stanley et al. 2016) ? Or are these patterns related to removal of CH4 from river water by emission to the atmosphere and by MOX within the delta, since the measurements were made quite away from the coast ?

I suggest that the authors make their data-set publically available, either as a supplement of the paper, or in an international data-base (PANGEA, MEMENTO, ...).

SPECIFIC COMMENTS

All of the abbreviations need to be defined, e.g. qPCR (L13), MISA (L14), OTUs (L21), etc...

L24-26: In estuaries there are typically differences in residence time in different regions (e.g. salinity ranges). Residence time will strongly affect the distribution of microbes that for some groups can have relatively long growth times.

L33: Please add a reference to back this statement on latitudinal variations of CH4 source-sinks.

L50: Conversely, the authors should also describe what goes on at depths <200 m since this corresponds to the regions covered by the paper.

L91: how was equilibration achieved ? Shaking ?

L101: Please add the reproducibility of peak areas of the standards, and the reproducibility of sample duplicates.

L 178: this equation was not given by Wanninkhof et 2009, it goes back at least to Liss & Slater (1974)

L226: Please add all of the station numbers to figure 1.

L232: I suggest that authors show the figures of the correlations as supplemental figures, in addition to the statistics in the Tables. The visual inspection of correlations can also be informative and useful.

L243: Please use nmol L-1 instead of nM throughout the text

L294: does the difference of 0.05 ppm in air CH4 have a significant incidence of the air-sea CH4 flux computation, given that the analytical uncertainty on the dissolved CH4 concentration is typically of +/- 3% ?

L 311: Can you provide a statistical test ?

L311: "a bit more north", can you quantify this in km ?

L318: I suggest to remove "unfortunately" this is a self-evaluation, let the reader decide what's unfortunate or not.

L335: "In contrast to sea-ice, the freezing and melting of freshwater-ice does not alter the salinity pattern": Please develop and clarify this statement, as I do not understand it. Melting of fresh-water ice and mixing with sea-water leads to a decrease of the initial salinity.

L340: then

L344: same as L318

Figure 2: please add a legend for the variable (and units) in the plot.

Figure 3: please add a legend for the variable (and units) in the plot. Add units in the text of the legend of the figure. It could be useful to add a plot with the horizontal distribution of salinity.

Figure 4: please add a legend for the variable (and units) in the plot. Add units in the text of the legend of the figure. This figure could be merged with Figure 2. It could also be useful to add the O2 vertical distribution along this transect.

Figure 5: legend of the figure is incomplete. Add the spatial (where) and temporal (when) info. The sediment data should also be in nmol/L. Add statistics of the regression. Please specify that the two crossed dots were excluded (I assume). Do you have an explanation why those two points are outliers ?

Figure 6: please add a legend for the variable (and units) in the plot. Add units in the text of the legend of the figure

Figure 7: please add a legend for the variable (and units) in the plot. Add units in the text of the legend of the figure

Table 2: How can r2 be negative ? Is this r ?

Table 2: what do the empty cases in the Table mean ? statistics not significant ? Please provide all of the stats and put in bold those that are significant.

Table 5: Specify this is for high latitude shelf seas.

REFERENCES

Fenwick, L., D. Capelle, E. Damm, S. Zimmermann, W. J. Williams, S. Vagle, and P. D. Tortell (2017), Methane and nitrous oxide distributions across the North American Arctic Ocean during summer, 2015, J. Geophys. Res. Oceans, 122, doi:10.1002/2016JC012493.

Liss, P. S. & Slater, P. G. Flux of gases across the air-sea interface. Nature 247, 181-184 (1974).

Stanley EH, Casson NJ, Christel ST, Crawford JT, Loken LC, Oliver SK. 2016. The ecology of methane in streams and rivers: Patterns, controls, and global significance. Ecological Monographs 86: 146–171.

---

## Author Comment (AC1) · 6 Jun 2017

**Reviewer 1**

This study highlights the minimal consumption of methane as a fraction of the dissolved gas, which is in flux to the atmosphere. Moreover, the authors demonstrated that in riverine, mixed, and polar water masses, MOX is significantly tied to methane concentration. The focus here is on the diffusive flux to the atmosphere, but we have no sense of how this diffusive flux compares with ebullition of methane from seeps in the study region. Since this study examined shallow water masses, discussion of any active seep/vent locations in the study area would be helpful, as ebullition is likely to play a major role in methane flux to the atmosphere, and, in turn what fraction of total methane release is available for consumption by MOB.

I am generally supportive of the publication of this study, although mention of marginally significant statistical findings or insignificant results and speculation leading from these should be addressed. In a few cases, grammatical errors and vague language should be rephrased, but addressing these items shouldn't be difficult.

*There is already a discussion on the effect on ebullition in lines 50ff, but we added some more points .....*

*L54: For lakes, it has been estimated that ebullition contributed to 18-22% of the total emission (Del Sontro et al. 2016).*

*L458: Ebullition of methane from the sediment in this area is also reported, resulting in very high methane fluxes 1 – 2 orders of magnitude higher than the other calculations (Table 3). The methane released by ebulltion did not show any isotopic evidence of oxidation and thus will be released almost completely into the atmopshere (Sapart et al. 2017). However, if this ebullition really results in elevated atmospheric methane concentrations is a matter*

Specific Comments:

L15 – here "methane distribution" refers in parentheses to "headspace", but this isn't a method and it is unclear what is meant. Suggest rewording.

*To our knowledge the measuring of methane concentration in a head space does represent a well-known method, we therefor reworded this to the methane distribution (via head-space method) and*

L44 – should read "The source(s) of methane..."

*Changed accordingly*

L55 – suggest rewording "water column MOX" to be consistent with first reference to an abbreviation (i.e. "water column methane oxidation (MOX)").

*Changed accordingly*

L59 – this sentence seems vague and perhaps unnecessary. *We prefer to keep thist statement*
Suggest beginning withthe following sentence and changing "for some authors" to "In certain studies"
*Changed accordingly*

L120 & L132 – change to methane [mono]oxygenase
*Changed accordingly*

L133 – were the same primers used here as above?
*Yes, changed accordingly*

L224-225 – "This was most pronounced..." the sentence is oddly phrased; suggest rewording.
*Changed to "This decrease off the coast was most distinct for the Transect 1 and 4, where also the maximal concentrations (218 nM) were observed".*

L230 – 236 "significant" should have a p-value given
*The p value is now added to the text.*

L286 – remove mention of OTU "preference" for different water masses, especially where you didn't find a significant trend. Perhaps use phrasing "association" or "link" instead of "preference" throughout.
*Changed to „association"*

L379-381 Perhaps MOB with divergent pmoA sequences were not detected with these specific primers? This possibility isn't discussed, but instead speculation was raised that MOB might exist that lack pmoA genes.
*We agree that our wording was not precise. We re-phrased the MS as follows: This could be due to the fact that there are MOB which were probably not amplified. The primer set used in this study is the most frequently used, however a couple of different primer sets are available for amplification of specific monooxygenase genes in several subgroups, which are not targeted using this primer set (Knief, 2015). Thus, these subgroups e.g. Verrucomicrobia or the anaerobic methanotrophic bacteria of the NC10 phylum and others (Knief, 2015) were not quantified in our study.*

L395-396 The statement that "OTUs identified in this study cannot be related to known MOBs" appears to contradict the taxonomic affiliations offered on Line 288. Do you mean that a subset of the OTUs identified in this study cannot be linked to known MOBs?

*Yes this is correct we re-phrased the MS accordingly*

L415-416 This part is a reiteration of the results on L295. What is the importance of measuring a higher windspeed in comparison to Thornton et al.?

*Changed to „This is a bit lower than 1.879 for the outer ice free Laptev Sea in summer 2014 as reported from Thornton et al., (2016). In contrast, our wind speed was a bit higher (4.2 ± 2.2 m/s) than 2.9 ± 1.9 m/s as reported from Thornton et al., (2016).. This would result in slightly higher equilibrium concentrations and higher gas exchange coefficient in our study"*

L443 Define (spell out) ESAS; not mentioned elsewhere.

*Changed accordingly*

Figure 3. I recommend changing the color for highest methane concentration from pale orange to something that isn't already on your color scale for lower concentrations (e.g., grey or black)

*I have dived into the program settings, but there seem to be no way to modify the range of colors.*

Figure 5. The omission of two data points is mentioned in the main text, but this should also be clearly stated within the figure caption.

*Changed accordingly*

---

## Author Comment (AC2) · 6 Jun 2017

**Review BGD** *Methane distribution and oxidation around the Lena Delta in summer 2013* **by Bussmann et al.**

Bussmann et al. present data from a measurement campaign in September 2013 in the coastal area close to the Lena river delta where river water and polar water mix. The activity (qPCR) and the abundance of methanotrophic bacteria was investigated and statistically compared to methane concentrations and physico-chemical parameters in order to determine environmental controls of MOX. Three water masses (river, mixed and polar) were defined previous to statistical analyses. This manuscript employs primers developed by Tavormina et al., which were even improved since the last publications by these authors. The use of these primers to investigate the methanotrophic marine community is quite new and I think that this is the strongest point of this manuscript. Conventional primers often don't cover the marine diversity. I enjoyed reading the manuscript since it is clearly written and everything is well-explained and a wide-range of literature is being put in context with the results of the presented study. There are, however, quite a few formatting/language mistakes. More importantly, I'm missing a more conclusive discussion (see below). If the remarks below can be addressed, most importantly the discussion, this manuscripts presents a solid addition to the current scientific pool of MOX studies and is suitable for publication in BG.

**General remarks:**
1) Did you try to analyze the data statistically without grouping it into different water masses? What are the results then? Or maybe set the salinity borders differently?
*Yes, we worked also with the whole data set, but no clear patterns were descernible then. We also applied the salinity border of Goncalves et al (at the same study site), but clearest results were obtained with the classification of Caspers. Also with North Sea data this was the "best" classification.*

2) It would be interesting to do qPCR with sediments samples from the river and coastal area. Especially for the 'outlier station' where authors hypothesize that part of the community got resuspended due to stormy weather. Was this done?
*Unfortunately we did not extract DNA from the sediment, eventhough it would have been important and very interesting.....*

3) The discussion is quite descriptive. I'm missing a more in-depth analysis of the results. For example, the third paragraph of 4.2 is very descriptive. What are the possible reasons that these communities are limited by different factors? Why is the riverine community more diverse? Due to stability? My opinion is that for the MS to be published in BG a less descriptive Discussion part is crucial.
We added the following paragraph to the section 4.2:
*Methane concentration and nitrogen availability are strong driving forces shaping MOB community composition and activity (Ho et al., 2013). Furthermore the interactions with other heterotrophic bacteria influence the methanotrophic community (Ho et al., 2014). As DOM removal and degradation occurs mainly at the surface / riverine water (Gonçalves-Araujo et al., 2015); this may also lead to an enriched methanotrophic population in the riverine water. We also assume that the riverine environment is exposed to more environmental changes (salinity, light), temperature) than the polar one. Changes in salinity have different impact on sensitive and non-sensitive MOBs, thus also shaping the methanotrophic community (Osudar et al., in revision). In contrast to our more divers riverine population, the methanotorphic population in the proper Lena river was*

*characterized by a rather homogenous community (Osudar et al., 2016). However, the classical concept of r- and k-strategist nowadays has been replaced by the C-S-R functional classification framework and type Ia MOB, responding rapidly to substrate availability and being the predominantly active community in many environments can thus be classified as competitors (C) and competitors-ruderals (C-R) (Ho et al., 2013).*

4) A wide range of statistical data is presented. It would be better to discuss the most important findings to avoid confusion of the reader.
*We moved 2 tables with statistical details to the appendix, and hope to make the text clearer.*

**Several small remarks, also with regard to formatting/language mistakes:**
-please check upper/lower case of chemical formulas/mathematical formulas
*We checked the text again and hopefully have now found all errors.*

-abstract line 11: biological "way" sounds a bit strange. Maybe biological sink?
*Changed accordingly*

-abstract line 21: riverine, not rivine
*Changed accordingly*

-abstract, line 22: "..riverine water TO (not AND).."
*Changed accordingly*

-abstract line 17: "..a median OF 28 nM.."
*Changed accordingly*

-line 44: hydrate not hydrated
*Changed accordingly*

-several times you write 'according to/XX to (XX et al, 1998)'. Please put the parentheses at the right place.
*We checked the text again and hopfully have now found all errors.*

-2.2 why are you using different chemicals (H2SO4 and NaOH) to kill samples for methane analyses for sediment and water samples.
*When measuring MOX the control values were lowest when applying H2SO4 to the water samples, thus we used the acid for all water samples. For sediment samples we used NaOH to avoid dissolution of any carbonate and subsequent CO2-production.*

-if you're sampling sediments with a grab sampler for methane analyses, is there not a lot of methane lost on the way up to the ship?
*The study area is very shallow, max. depth 20 m, thus the grab sampler took only few minutse to return on board.*

-line 199: remove the 'than'
*Changed accordingly*

-line 238: herEby
*Changed accordingly*

-if you're correlating MOX to CH4: how can you be sure that's possible since MOX=CH4*k. Isn't what you're calculating then just assessing if k is much smaller than the CH4 concentration (which it generally is).
*Yes, we are aware that this corelation is "difficult", because of this autocorrelation. Nevertheless, it is often used in the literature and the differences between the 3 groups are very strong. We added the following sentence "However as MOX is calculated with the methane concentration, this correlation has to be regarded with caution."*

-line 311: "..seemed to be.."? or there was none?
*Changed to "there was no significant difference"*

-line 324: degradation processes? You mean methanogenesis in the sediments?
*Yes, changed to "This correlation can be related to degradation processes finally leading to methanogenesis,... "*

-line 334 and after: I can't really follow your explanations. Could you rephrase/shorten/write it clearer. I might have missed something but I did not get your point.
*We try to explain the missing correlation between freshwater input from the river and the methane concentration. If there is another freshwater source (from ice melting) with low methane concentrations (in contrast to the riverine freshwater with high methane content) this could explain the missing correlation. We re-phrased the paragraph to make it clearer.*

-4.2: there was recently a paper published in BG about MOX in coastal environments (Baltic Sea, Eckernförde Bay). Would be good to include it.
*This work in now included.*

-line 356: "..fractional turnover rateS.."
*Changed accordingly*

-line 375: "..but more..": what do you mean? More than no correlation? Please rewrite.
*Changed to "but correlations to ....."*

-line 380: what's the different from dormant MOB to not active MOB? Do you mean dormant, for instance as endospores? Please write more clearly. Like this, it reads like a repetition from line 376.
*Yes, it is a sort of repetition, but the first (in line 376) is a general statement concerning the restricion of the method, and the line 380 refers to more specifically to methanotrophic bacteria.*

-line 403 and 407: limited or influenced? I would prefer a clearer way of writing this.
*Ok, they were limited (negative correlation)*

-line 433: where was Graves et al., 2015 measuring fluxes?
*They calculated the methane flux, as the other studies in this sentence.*

-line 437: did Sapart et al. not measure atmospheric fluxes? Graves et al., 2015 also measured atmospheric methane.
*Yes, they also measured the atmospheric concentrations, but the flux was calculated*

*based on the water borne methane concentrations (bottom up). In contrast to Myrhte and Thornton, whose flux calculations were based on the atmospheric concentrations (top-down).*
*We changed the sentence to … few studies focus on the atmospheric concentrations...."*

-line 439: remove the ":"
*Changed accordingly*

-line 443: what is ESAS?
*East Siberian Arctic shelf (ESAS)*

-line 447: change than to then (also at other places in the MS, please double-check)
*Changed accordingly and throughout the text*

-line 451: there was recently a paper published in BG about MOX in coastal environments (Baltic Sea, Eckernförde Bay). Might be interesting to compare the two.
*A comparison is now included in the text, L463 ff*

-Figures made with Ocean Data View: Make sampling spots more visible! It would be better not to use the mode where two data-points merge together (interpolation) since there are so few data points.
*The stations are now indicated with a black dot within the colored circles (Fig. 3, 6 and 7), in figure 2 the stations are indicated with a vertical line.*

-Figures: check lower/uppercase
*Changed accordingly*

-Table 5: there is not a very good coverage for shelf seas (eg North-Am. Coast, Baltic Sea)! I enjoy this table and it would be good to extend it a bit.
*The Baltic Sea and the North Am Coast are now included!*

---

## Author Comment (AC3) · 6 Jun 2017

GENERAL COMMENTS

Bussmann and colleagues report a valuable data-set of dissolved CH4 concentration in the Lena Delta.

It could be useful if authors compare in much more detail their new data-set with older data-sets obtained in the area (Bussmann et al. 2013). As it stands it's unclear what's the added value and novelty of the present ms compared to what was previously published by the authors on the same topic.

*In the present study only transect 1 overlaps with the previous study, most of the present sites are more to the north. As a novelty of this study we also assessed the influence of methane oxidation on the methane distribution pattern. As specified at the end of the introdcution: "The aim of this study was to get an overview of the methane distribution in the near shore and northern parts of the Laptev Sea and to gain insight into the role of methane oxidizing bacteria in the methane cycle in this area. Furthermore we tried to assess which environmental factors determine the methane distribution and its oxidation".*

The CH4 concentrations in the study area are extremely low compared to other estuarine environments (at lower latitudes), and the spatial gradients are also extremely low given the large salinity gradients. This fundamental difference contains some potentially important information on the functioning of estuaries in high latitudes and deserves to be discussed in light of published CH4 data in other estuaries. Is this due to a low CH4 concentrations in the Lena inner river itself? Any data on the CH4 concentration in the river itself ? If so does it differ from other rivers worldwide (e.g. Stanley et al. 2016) ? Or are these patterns related to removal of CH4 from river water by emission to the atmosphere and by MOX within the delta, since the measurements were made quite away from the coast ?

*The following sentence is now added to the discussion 4.1: "Methane concentrations in the Lena River, Bykowski Channel are on average 58 ± 19 nM (Bussmann 2013 and unpublished data from 2012 and 2016). This is much lower than the average global riverine methane concentration of 1350 ± 5160 nM [Stanley, 2016 #2645]. However, for the esturies of the Ob and Yenisei similar low concentrations are reported; 18 ± 16 nM from [Savvichev, 2010 #2447] and approx. 30 nM from [Kodina, 2008 #2485]."*

I suggest that the authors make their data-set publically available, either as a supple- ment of the paper, or in an international data-base (PANGEA, MEMENTO, . . .).

*The methane related data set is already available at www.pangaea.de, doi:10.1594/PANGAEA.868494, 2016. This is now stated in Line 103 and L481*

SPECIFIC COMMENTS

All of the abbreviations need to be defined, e.g. qPCR (L13), MISA (L14), OTUs (L21), etc*. . .*

*We agree with the reviewer, however the whole definition of these methods would be rather long. Thus we suggest that the interested reader should refer to the M&M section and we would rather keep the abbreviations in the abstract.*

L24-26: In estuaries there are typically differences in residence time in different regions (e.g. salinity ranges). Residence time will strongly affect the distribution of microbes that for some groups can have relatively long growth times.

*We added the following sentence to the discuccion 4.4: "In estuaries the residence time of the water (as influenced by water discharge and tidal force) also influences the effidiency of the estuarine filter (Bauer et al., 2013)."*

L33: Please add a reference to back this statement on latitudinal variations of CH4 source-sinks.

*We refer now to Saunois et al., 2016.*

L50: Conversely, the authors should also describe what goes on at depths <200 m since this corresponds to the regions covered by the paper.

*The next sentence does refer to water < 200 m: "However, ebullition at shallow water depths represents a short cut as it will not dissolve into the water, and most of this methane will reach the atmosphere. For lakes, it has been estimated that ebullition contributed to 18-22% of the total emission (Del Sontro et al. 2016)"*

L91: how was equilibration achieved ? Shaking ?

*Yes, the following is added to the text now: "The samples were vigorously shaken and equilibrated for at least two hours".*

L101: Please add the reproducibility of peak areas of the standards, and the repro- ducibility of sample duplicates.

*The precision of the calibration line was r^2 = 0,99, the reproducibility of the samples 7%. This information is now added to the M&M section, 2.2*

L 178: this equation was not given by Wanninkhof et 2009, it goes back at least to Liss

& Slater (1974).
*Corrected accordingly*

L226: Please add all of the station numbers to figure 1.
*Changed accordingly*

L232: I suggest that authors show the figures of the correlations as supplemental figures, in addition to the statistics in the Tables. The visual inspection of correlations can also be informative and useful.
*Reviewer 2 "complained" about to much statistics, thus we think that showing only the tables is a good compromise giving all the essential informations.*

L243: Please use nmol L-1 instead of nM throughout the text
*Changed accordingly*

L294: does the difference of 0.05 ppm in air $CH_4$ have a significant incidence of the air-sea $CH_4$ flux computation, given that the analytical uncertainty on the dissolved $CH_4$ concentration is typically of +/- 3% ?
*Well, the reviewer is right here, however these are the numbers as given in the data base.*

L 311: Can you provide a statistical test ?
*Has been changed to: "Overall, there was no significant difference (Wilcoxon Rank Sign Test for paired data, n = 18, p = 0.84)".*

L311: "a bit more north", can you quantify this in km ?
*No, the figure in this publication does not give enough details, thus it is changed to "In the same study area and in summer 2014"*

L318: I suggest to remove "unfortunately" this is a self-evaluation, let the reader decide what's unfortunate or not.
*Well, I think most readers will agree that missing data are "unfortunate", thus we would prefer not to change our statement here.*

L335: "In contrast to sea-ice, the freezing and melting of freshwater-ice does not alter the salinity pattern": Please develop and clarify this statement, as I do not understand it. Melting of fresh-water ice and mixing with sea-water leads to a decrease of the initial salinity.
*We modified the paragraph to:*
*"One reason could be another source of freshwater, but with low methane concentrations. In contrast to other estuaries, arctic estuaries are ice covered about 2/3 of the year and the seasonal freezing and melting of ice has a strong impact on the water budget. The freezing of sea water results in brine formation with strongly increased salinity, while its melting results in a freshwater input (Eicken et al., 2005). In contrast to sea-ice, the freezing and melting of freshwater-ice does not alter the salinity pattern. In 1999, the river water fraction in ice-cores near our study area ranged from 57% - 88% (Eicken et al., 2005), thus at least some additional non-river-freshwater input is possible. Even though not much is known about methane concentrations in ice, based on a recent study on sea-ice in the East Siberian Sea (Damm et al., 2015), we assume that this melt water probably has lower methane concentrations than the river-freshwater. This additional aspect of the water budget in ice covered eaturies might*

*explain the missing relation between salinity and methane concentration. "*

L340: then
*Changed accordingly*

L344: same as L318
*Well, I think most readers will agree that missing data are "unfortunate", thus we would prefer not to change our statement here.*

Figure 2: please add a legend for the variable (and units) in the plot.
*Changed accordingly*

Figure 3: please add a legend for the variable (and units) in the plot. Add units in the text of the legend of the figure. It could be useful to add a plot with the horizontal distribution of salinity.
*The units are now added. The salinity is shown in a supplementary Figure A2*

Figure 4: please add a legend for the variable (and units) in the plot. Add units in the text of the legend of the figure. This figure could be merged with Figure 2. It could also be useful to add the O2 vertical distribution along this transect.
*The units are now added to the figure and the legend. Figure 2 and 4 are now merged to figure 2a and 2b. We checked on the O2 distribution, but it was rather uniform and we think it would not give additional insights.*

Figure 5: legend of the figure is incomplete. Add the spatial (where) and temporal (when) info. The sediment data should also be in nmol/L. Add statistics of the regression. Please specify that the two crossed dots were excluded (I assume). Do you have an explanation why those two points are outliers ?
*The sediment methane concentrations have been modified and the legend modified to: "Correlation between the methane concentration in bottom water and the concentration in the underlying sediment for all stations (r2 = 0.62, p < 0.001, n= 33) . Two very high values from station TIII-1304 were excluded from the analysis. "*
*The high concentrations at station TIII-1304 are discussed in paragraph 4.1*

Figure 6: please add a legend for the variable (and units) in the plot. Add units in the text of the legend of the figure.
*The legend is the plot is now modified and the units are explained in the figure legend.*

Figure 7: please add a legend for the variable (and units) in the plot. Add units in the text of the legend of the figure
*The legend is the plot is now modified and the units are explained in the figure legend.*

Table 2: How can r2 be negative ? Is this r ?
*Ok, the negative sign should indicate a negative correlation, thus we put the "-" in brackets.*

Table 2: what do the empty cases in the Table mean ? statistics not significant ? Please provide all of the stats and put in bold those that are significant.
*All statistics are now provided, however in response to reviewer 2 we have moved the tables to the supplementary material.*

Table 5: Specify this is for high latitude shelf seas.

*As referee requested a reference from a boreal bay, we do not think this addition is justified.*

REFERENCES

Fenwick, L., D. Capelle, E. Damm, S. Zimmermann, W. J. Williams, S. Vagle, and P. D. Tortell (2017), Methane and nitrous oxide distributions across the North American Arctic Ocean during summer, 2015, J. Geophys. Res. Oceans, 122, doi:10.1002/2016JC012493.

Liss, P. S. & Slater, P. G. Flux of gases across the air-sea interface. Nature 247, 181- 184 (1974).

Stanley EH, Casson NJ, Christel ST, Crawford JT, Loken LC, Oliver SK. 2016. The ecology of methane in streams and rivers: Patterns, controls, and global significance. Ecological Monographs 86: 146–171.

---

## Editor Decision (ED1)

BG-2017-22

Dear Dr Bussmann

Thank you for the revision of your MS and your letter that was forwarded to me by the editorial manager. From my reading of this new version, I acknowledge the important changes in the MS that considerably improve the scientific argumentation. I think your MS has reached the overall scientific quality to be published in Biogeosciences, and this is why I will recommend publication. However, the publication can occur only after some more serious revision of the language. Indeed, the poor editing quality of your MS again disappointed me, and I find it hard to believe that it has been edited by a professional service as you mention in your letter (otherwise, I would recommend moving to another editing service). Like you, I am not English native speaker, and I understand it is not easy to publish in English. However, I cannot accept that the poor quality of language alters the scientific message. The problem is that at many places in your MS (see list below), the text seems more spoken language than written language. The scientific meaning becomes unclear and/or imprecise. A detailed language editing is absolutely necessary to make scientific statements unambiguous before publication.

I would like to point out that, contrary to what you mention in you letter, anonymous referee #3 was the same person during the first and second round of the review process. Her/his evaluation was much more severe the second time, because she/he considered you did not satisfactorily and seriously address her/his comments. Please keep in mind that reviewing is a volunteer time-consuming work. As associate editor, I ask you to consider my comments more seriously this time. Below, some citations from your MS that suggest it has not been edited by a professional service as you pretend. Again, I am not English native, so this list is probably not exhaustive and the entire MS needs language editing.

Looking forward a revised version of your MS,
With best regards

Gwenaël Abril, BG associate editor

L24 (and throughout the MS): "a higher "estimated diversity" in the "riverine water" than in the "polar water"." Ambiguous use of quotation marks; quotation marks can be use once a time when defining a term that will appear later in the MS, but not throughout the MS.

L46 "global runoff" : change to "global river runoff"
L61: "an important effect on reducing the greenhouse effect considerably" > please rephrase

L62  "the final sink for methane before it is released to the atmosphere" awkward sentence, what is the meaning of "final" here?

L95 "as modified from (Caspers, 1959)." Please refer to how to cite literature and use parentheses (same comment at many places in the MS)

107 "were calculated by differential weighing" >I suggest: "were determined gravimetrically"

L214: change "square" to "rectangle"

L234: "warm freshwater at the surface (0–5 m), followed by a mixed water body" not clear what "followed" means here... below? Above?

L235 "we found cold and saline water (= "polar water")" please make a sentence

L238 "the subsequent stations" unclear, please specify

L248 "When applying our water masses (riverine, mixed and polar), we observed..." awkward sentence

L257 ""riverine" sample) and excluding the very high water values of station TIII-1304; which made the correlation much stronger" Ambiguous use of quotation marks and semicolon. What made the correlation stronger, the high value or to exclude the high value?

L272 "The fractional turnover (k') is a measure of the relative activity of the MOBs" > how was k' calculated?

L279 "The abundance of MOB can either be given either in cell numbers or as relative abundance" awkward sentence

L290 "Additionally, the "estimated diversity", as OTUs per station," ambiguous use of quotation marks

L334 "reported a range of 10 to 100 nmol L-1 (estimated from Figure 2 in (Sapart et al., 2017)." inappropriate use of parentheses (also at many other places in the MS)

L339: "A more detailed comparison with temperate and tropical environments is discussed below, in the context of the diffusive methane flux, as most reviews rely on the methane emissions rather than on the concentrations." What reviews? Please add references

L345 "seafloor methanogenesis resulting from the decomposition of organic carbon" this is a truism as methanogenesis IS decomposition of OC

L347 "However, low tides, low topographic relief and low precipitation in the present study area are not favourable for a high ground water input to the Lena Delta." Low tides: do you mean low tidal amplitude? In addition explain the scientific reason for this statements.

L353 "Thus, we cannot support ebullition as a methane source." awkward sentence, please check

L365 "However, for the latter two processes, it not clear yet if this methane production will result in elevated methane concentrations in situ." Please rephrase

L368 "However, at the shallow stations (< 8 m, coastal stations of the transects), the water column was mixed; thus, sedimentary methane may diffuse into the water above." Unclear, and oversimplified statement, please rephrase.

L372 "as rivers or estuaries are thought to import methane-rich water into coastal seas." Do you mean "export" instead of "import"?

L378 "However, even this scheme does not seem applicable to our data." Please improve formulation to be more explicit

L393: "median methane oxidation rate of 0.32 nmol L-1 d-1, ranging from 0.03 to 5.7." please avoid numbers without unit

L401 "The first order rate constant used for modelling the methane flux in the Laptev Sea is estimated to range from 18116 d-1 to 11 d-1 (= 2.3 × 10-6 to 3.8 × 10-3 h-1)" Confusing: was this rate "estimated" or "used"; avoid "=" in the text.

L407 "The influence of the methane concentration on the MOX was also most pronounced in "riverine water"" inappropriate use of quotation marks

L409 "With the described method of qPCR and the water column specific primers (Tavormina et al., 2008), the relative abundance of MOB in our study ranged from 0.05 to 0.47% (median 0.16%) which is equivalent to $4 \times 10^4$ to $3 \times 10^6$ cells per L (median of $6.3 \times 10^4$), except for the high values from station T1-1302." Confusing. What is the high value ?

L412: "; thus, they are regarded as methodological outliers" awkward sentence, please justify how such methodological problem may occur.

L422 "correlations between parameters important to heterotrophic bacteria," please rephrase, bacteria don't mind about "parameters"

L451 "Thus, the ecological traits can be described as follows:" not sure "thus" is appropriate at the beginning of a new paragraph. I suggest changing "described" to "summarized"

L454 "The relative abundance and "estimated diversity" (OTU/sample) of MOB" inappropriate use of quotation marks

L456 "the MOB in the polar population were quite efficient at reaching a MOX comparable to riverine water." Do you mean MOB population in polar waters? do you mean MOX activity, or MOX rates? do you mean comparable to MOX activity of MOB in riverine waters? Please rephrase

L469 "the classical concept of the r- and k-strategist has today been replaced by the C-S-R functional classification framework" please explain

L491 "ranging from 4–163 μmol m2 d-1." Ranging from 4 to 163 μmol m2 d-1

L494 "A comprehensive study by (Myhre et al., 2016)" inappropriate use of parentheses

L498 "(Graves et al., 2015) (Table 5Table 5)." Please edit your MS

L509 "A comprehensive study by (Myhre et al., 2016)" inappropriate use of parentheses

L509 "aquatic methane emissions is presented by (Stanley et al., 2016) and (Ortiz-Llorente and Alvarez-Cobelas, 2012). inappropriate use of parentheses. Same in many other places in the MS

L512 "as well as from tidal systems, to which the Lena Delta would be classified" "to which" is not correct

L514 "; thus they conclude" to what do "they" refer to ?

L520 and at many other places in the MS "(Borges et al., 2017;Lofton et al., 2014)" insert a space after the semicolon

L525 "A molecular approach identified the salinity, temperature and pH as the most important environmental drivers of methanogenic community composition on a global scale." Please add a reference

L534 a paragraph cannot start with ")."

L534 "he methane released by ebullition did not show any isotopic evidence of oxidation;" please provide a reference

L556 "However, one fact to be kept in mind is that our estimation is a static one" poor English

L561 "A more complex approach is taken by (Wahlström and Meier, 2014)." Poor English and inappropriate use of parentheses

L576 "as no direct dilution of riverine methane occurs;" awkward sentence, how can "dilution" be "direct" (or "indirect")?

L583 "; thus, we propose that it will compensate for any increase in methane concentrations." Not clear, is this speculation? Do you mean you "postulate"?

L585 "he stratification of the water column will be broken up and the separate water masses mixed" looks like spoken English, not written English.

Managing Director Thies Martin Rasmussen

\*\*\*\*\*\*\*\*\*\*\*\*\*\*\*\*\*\*\*\*\*\*\*\*\*\*\*\*\*\*\*\*\*\*\*\*\*\*\*\*\*\*\*\*\*\*\*\*\*\*\*\*\*\*

From: Ingeborg Bussmann [mailto:Ingeborg.Bussmann@awi.de]
Sent: 28 Aug 2017 15:50
To: Copernicus Publications Editorial Support
Subject: Re: bg-2017-22 (author) - manuscript files under validation

Dear Natascha,

I wanted to add some words to the editor, but pressed the wrong button….

Could you forward this to the editor?

yours sincerely

Ingeborg Bussmann

Dear Gwenael Abril,

Thank you for the valuable suggestions from you and referee 4.

The Ms now has been edited by professional service and the Ms is now better to read (at least for my feeling). The discussion on methane concentrations and methane fluxes have been completely re-written to a hopefully more coherent way, but without changing the message. In addition, in the conclusion we relate future warming in the Arctic to the methane cycle in the study area based on our discussion.

We appreciate the comments of the additional referee 4, however we also think they are rather harsh, as the 3 previous referees were more positive on the Ms and we already had incorporated their suggestions.

Nevertheless, we worked again on the discussion, polished the figures and the language and hope the Ms has improved to your satisfaction.

Yours sincerely

Ingeborg Bussmann

---

## Author Response (AR2)

**COMMENTS TO THE EDITOR**

**Associate Editor Decision: Reconsider after major revisions** (19 Jul 2017) by Gwenaël Abril
Comments to the Author:
Decision #3 on bg-2017-22 Methane distribution and oxidation around the Lena Delta in summer 2013

Dear Authors
I have received a second review of anonymous referee #3, and her/his evaluation of your revised MS was quite severe (see report). Note that this reviewer is an expert in the field. I made my own reading of your revised MS, including the modifications in the first version, and, indeed, I agree with referee#3 that your revisions were too superficial and not sufficient to satisfy all initial comments and reach the standards of a high quality journal like Biogeosciences. However, I believe this work must be published in BG, but only after a more careful and detailed revision. Critics concern:

(1) presentation of the MS and formatting of figures and tables, and legends;
*The Ms has now been checked by a professional editing service; and the figures have been polished.*

(2) a too superficial discussion that does not adequately refer to the literature in order to put the presented results in a broader context;
*The discussion on methane concentrations and methane emission, as well as the conclusion have been completely rewritten. Relating the data also to worldwide data sets, and extrapolating our data to the warming of the Arctic climate.*

(3) some imprecise or inappropriate statements in the interpretation of the data.
*We checked the whole Ms again, as well as the editorial service, and think the text has much improved now.*

I would be pleased if you could revise your MS in order to satisfy the referee's comments (and mine you will find below), and provide a detailed list of responses and a description of what as been changed in the MS.
Looking forward to reading this soon.
Best Regards
Gwenaël Abril, BG associate editor

Additional comments:
L204, what were the criteria used to delimit the two rectangular area. Appendix Fig. A1 is not necessary if the rectangles appear in Fig.1
*We tried to envelope the whole study area, which was best obtained with two rectangles which are bordered by the most southern,northern, eastern and western stations. This area is now indicated as dashed line in the map.*
*The text is now changed to:*
*"Two rectangles which are bordered by the most southern, northern, eastern and western stations gave a good estimation of investigated area Figure 1."*

L234 and throughout MS: "pale orange" is inappropriate as in fact it is "pale pink", pale orange appearing between the yellow and the red in the colour grid. All the figures drawn with ocean data view are of poor quality (coloured points are very large and sometimes overlap) and the mentions "@depth(m)=last" are not understandable. Please improve also the legends.
*We have polished now all odv plots, including the legends. When some data points are above the range, their number is now indicated with their real number.*

Notation „riverine water" is not conventional and unacceptable
*But the word is in the Oxford Dictionary, with" riverine = Relating to or situated on a river or riverbank; riparian". The professional editing service also had no objections......*

I believe the information in the appendix tables 1 and 2 could appear together in a same table in the MS.
*Ok I have incorporated them again into the text, even though referee 3 wanted these table in the supplement.....*

Why didn't you test the correlation of CH4 concentration and MOX activity only after separating "river" "mixed" and "artic" and for the whole dataset?
*Best results were obtained when splitting the dataset. And as riverine and polar water are separated by a strong pycnocline, we think it is more reasonable to do the statistics separately.*
*L 248 ff*
*"When applying our water masses (riverine, mixed and polar), we observed significantly different methane concentrations in these water masses, with medians of 22, 19 and 26 nmol $L^{-1}$ (p = 0.03), respectively(table1). Therefore, the linear correlation analysis was performed separately for the different water masses.!*

Please also consider a presentation of CH4 concentrations versus salinity, as this is very classical and broadly used in the literature.
*A new figure 8 with methane versus salinity is now added to the ms and "we thus conclude that the classical way of river water dilution does not apply for the Lena Delta". Line 373 ff*

L317 The question raised by the referee of the general low CH4 concentrations in artic rivers and estuaries compared to other regions in the world is to my opinion fundamental and deserves a specific paragraph in the discussion: why are CH4 concentrations lower in artic estuaries?
*A additional paragraph, comparing tropical, temperate and arctic waters has been added, lines L516 ff*

L344 Reader is lost here. CH4 vs salinity plots would help the discussion.
*The whole discussion of methane in surface water has been re-written, and we now also include a figure with methane versus salinity. Line 373 ff*

L346 "we thus exclude the Lena River as methane source" strong statement not based on quantitative analysis. In theory you should be able to calculate the CH4 input from rivers using concentration and discharge.
*Ok, the wording was misleading, we did not want to make up a budget calculation. Thus we only conclude that the concept of salinity versus methane is not applicable for our data set.*
*L 380 ff*

L347"One reason for this missing correlation, could be another source of freshwater, but with low methane concentrations." Formulated that way, it appears as a speculative statement.

L355 "This additional aspect of the water budget in ice covered estuaries might explain the missing relation between salinity and methane concentration." Ok, but please improve this part of the discussion (and associate figures) to make it clearer.
*New L 382, the paragraph has been reworded*

L456 "matter of debate between biogeochemists, ecologists, and PHYSICISTS" please revise English

*Corrected*

**Comments to Referee 4**

L8 : « biggest" in terms of what ? Discharge ? Drainage area ?
*Changed to "largest river" concerning discharge*

Please define the abbreviations (qPCR, MISA, etc…) in the abstract. The abstract should be understood on its own, which requires that abbreviations are defined. Refer to instructions "The abstract should be intelligible to the general reader without reference to the text", which is obviously not the case if the abstract is full of abbreviations. Since there is no size limit for abstracts in BG it should be not a problem to define all of the abbreviations.
*The abbreviations are now replaced by the full name of the method.*

L64: "MOBs" abbreviation is not defined, as it should the first time an abbreviation is used. Same applies to DOC, TDN, etc… The authors should take the time to polish the presentation of their work. It's not the reviewer's job to check this, especially at the second round of review.
*MOB has been explained above in line 59. The DOC and TDN are now explained. In addition the Ms has been now polished by a professional editing service!*

L74: Here, explain how the present paper adds to the previous papers by the authors in the area.
*In this study we focus more on the northern part of the Lena Delta and for the first time we were also able to measure the microbial methane consumption rate. New L76 "*The aim of the present study was to obtain an overview of the methane distribution in the northern parts of the Lena Delta and to gain the first key insights into the role of methane-oxidising bacteria (MOB) in the methane cycle occurring in this area*"*

L97: Specify what was the delay between the sampling and the analysis of CH4 in the home laboratory ? How were samples stored between sampling and analysis in the home lab: room temperature ? Did the authors check if poisoning with H2SO4 is efficient to stop biological activity and for how much time ? While I can imagine that acidification can preserve a sample for a few days, I not sure this is adequate for storage for months.
*Sample preservation is always a critical point. However, our stoppers are rather methane save and a strong shift in pH is preferable to the more difficult to handle and environmental dangerous HgCl2. With a similar set up, Magen et al could show that methane concentrations in preserved samples with methane concentrations > 1 ppm did not change over a year [Magen, 2014 #2514]. Even though his shift in pH was obtained by NaOH. Samples poisoned with acid and stored in glass bottles with butyl stoppers also did not change in DIC content [Taipale, 2009 #2791].*
*We changed the text to:*
*The samples were stored upside down at temperatures < 15°C and analyzed after 4 months. Glass bottles and butyl stoppers are relatively methane tight and acidification of water samples results in good long-time sample preservation[Magen, 2014 #2514; Taipale, 2009 #2791].  However, we cannot exclude that some methane of the samples was lost.*

L101: specify the concentration of the NaOH solution.
*It was 1 M NaOH*

L185: it's Liss not Lisa
*Sorry, for the misspelling, in the references it was correct…*

L 195: the relationship was not "developed for coastal seas", it was derived from an experiment in a continental shelf. In fact the Nightingale relationship converges with relationships derived in the Southern Ocean (Ho et al. 2006).
*Ok, changed to "obtained for coastal seas"*

L 235: "very high" is not adequate to describe CH4 concentrations since they vary by several orders of magnitude accross aquatic systems. 400 nmol/L might be "very high" for the Laptev Sea but will be very low for a tropical reservoir. Refer to actual numbers.
*The actual numbers are now given, and changed to ". At station TIII-1304 we also observed comparably high methane concentrations in surface (212 nmol L$^{-1}$, figure 3)*

L 236: use an uniform unit throughout the text, a not a mix of µM, nM, nmol/L.
*All concentrations are now given in nmol/L, only the fluxes are given in µmol/L/m2.*

L 321: Did the vertical profiles of T and S show a mixing in response to "wind increase" ?
*The water depth at this station was very shallow (4 m) and very "bumpy" at the time of sampling, thus no CTD cast is available, only information on the water of the Niskin bottle.*
*We also decided to remove the "outlier" TIII1304 from the discussion, but focus on the other stations.*
*Section 4.1 Methane concentration and 4.3 diffusive methane flux have been restructured in a hopefully more coherent way.*

L 327, 363: "Unfortunately" is a subjective evaluation. Scientific writing should neutral and objective.
*Ok, has been removed.*

L 333: Oxygen in surface waters of rivers/estuaries can also be lowered by sediment organic matter degradation, since sediment respiration is usually equivalent to water column respiration in estuaries (Gazeau et al. 2004). So the correlation of CH4 and O2 does not necessarily imply that CH4 is produced in the water column due to a process that consumes O2 in the water column.
*Yes, but as we have a strong pycnocline separating surface and bottom water, and as O2 is only influencing CH4 concentrations in the surface water, we still think that in situ methane production is a possible methane source. See new Line 355 ff*

L336: Is there evidence that DMSP occurs in the Lena river/estuaries ? The study of Florez-Leiva et al. 2013 shows that the increase of CH4 in response to a spike in DMS was 2 nmol/L/d. Could this production term sufficient to account for the CH4 concentrations ?
*The process of in situ methane production is now discussed in more detail, but also stating that it is not clear yet, whether the experimental shown process really will result in elevated in situ methane concentrations.*

L338-345: The CH4 concentrations in the Arctic estuaries (Lena, Ob and Yenisei) are much lower than those in temperate and tropical estuaries. This is an important information from this paper that should be mentioned and discussed. This probably results from the low temperatures and possibly lower organic matter concentrations in rivers (although this needs checking and discussing). This is relevant because temperature and possible organic matter concentrations in rivers can change in future in Arctic estuaries.
*The low CH4 concentrations and CH4 emissions are now discussed in relation to temperate and tropical estuaries (L 506ff) and in the conclusion we extrapolate our data to future changes in the Arctic.*

L 460: what is the reproducibility and accuracy of these measurements ? Is a 0.02 ppm difference significant, given differences in calibration gases, methods, etc… ?
*Probably, this is no real difference, thus we state that our values within the range of literature values. L 481.*

L 466: This not the case in the Southern part of the North Sea where there is no thermal stratification, and fluxes are much higher than in the Northern part of the North Sea where stratification occurs (Borges et al. 2016).
*This references is now added to the text and the comparative table. Also the influence of stratification is discussed L504 and L516*

L 520: diverse
*corrected*

Figure 2: remove PSU. Salinity is measured as a ratio of two conductivities (sample:standard), hence, unitless.
*corrected*

In the plot of Figure 2 replace nM by nmol L-1
*corrected*

In plot of Figure 3 replace nM by nmol L-1 and remove @ depth (m) = first
*corrected*

It could be useful to add the salinity plot (appendix figure) to figure 3, so that the CH4 concentration can be compared directly to the salinity (on the same figure).
*We preferred to add a new figure 7, showing the methane concentration versus salinity, as suggested by the editor.*

In plots of Figure 5 add the units of MOX, and remove @ depth (m) = first
*corrected*

Appendix Table A1: specify if this was done only for the surface data or the full data-set (all depths). Specify what do the bold numbers mean.
*The table legend has been changed to:*
*Appendix Table A1. Linear correlation between the methane concentration versus different environmental parameters splitted into three water masses with their whole respective data set. Analysis was performed with log transformed data, shown are the $r^2$-values, the level of significance (p) and the positive or negative correlation (+/-), bold numbers indicate a significant correlation (p<0.05).*

References
Borges AV, W Champenois, N Gypens, B Delille, J Harlay (2016) Massive marine methane emissions from near-shore shallow coastal areas, Scientific Reports, 6:27908, doi:10.1038/srep27908
*Has been included now*

Gazeau et al. (2004) The European coastal zone: characterization and first assessment of ecosystem metabolism, Estuarine, Coastal and Shelf Science 60, 673-694

Ho, D. T., C. S. Law, M. J. Smith, P. Schlosser, M. Harvey, and P. Hill (2006), Measurements of airsea gas exchange at high wind speeds in the Southern Ocean: Implications for global parameterizations, Geophys. Res. Lett., 33, L16611, doi:10.1029/2006GL026817.

---

## Author Response (AR3)

Dear Dr. Abril,

Thank you again for your comments.
The Ms has been sent again to the same editing service and they have polished the whole Ms and I also have checked the Ms again. I have outlined our answers to your specific comments in the text below.
I think the language has now improved significantly and I hope you will appreciate the changes.

Yours sincerely

Ingeborg Bussmann

BG---2017---22

Dear Dr Bussmann

Thank you for the revision of your MS and your letter that was forwarded to me by the editorial manager. From my reading of this new version, I acknowledge the important changes in the MS that considerably improve the scientific argumentation. I think your MS has reached the overall scientific quality to be published in Biogeosciences, and this is why I will recommend publication. However, the publication can occur only after some more serious revision of the language. Indeed, the poor editing quality of your MS again disappointed me, and I find it hard to believe that it has been edited by a professional service as you mention in your letter (otherwise, I would recommend moving to another editing service). Like you, I am not English native speaker, and I understand it is not easy to publish in English. However, I cannot accept that the poor quality of language alters the scientific message. The problem is that at many places in your MS (see list below), the text seems more spoken language than written language. The scientific meaning becomes unclear and/or imprecise. A detailed language editing is absolutely necessary to make scientific statements unambiguous before publication.

I would like to point out that, contrary to what you mention in you letter, anonymous referee #3 was the same person during the first and second round of the review process. Her/his evaluation was much more severe the second time, because she/he considered you did not satisfactorily and seriously address her/his comments. Please keep in mind that reviewing is a volunteer time---consuming work. As associate editor, I ask you to consider my comments more seriously this time. Below, some citations from your MS that suggest it has not been edited by a professional service as you pretend. Again, I am not English native, so this list is probably not exhaustive and the entire MS needs language editing.

Looking forward a revised version of your MS,
With best regards

Gwenaël Abril, BG associate editor

L24 (and throughout the MS): "a higher "estimated diversity" in the "riverine water" than in the "polar water"." Ambiguous use of quotation marks; quotation marks can be use once a time when defining a term that will appear later in the MS, but not throughout the MS.
*To my opinion, the quotation marks for names should be kept throughout the text, but probably I am wrong here and I have removed them.*

L46 "global runoff" : change to "global river runoff"
*Corrected*

L61: "an important effect on reducing the greenhouse effect considerably" > please rephrase
*Changed to "an important impact on reducing the greenhouse effect considerably"*

L62 "the final sink for methane before it is released to the atmosphere" awkward sentence, what is the meaning of "final" here?
*Changed to "therefore the final sink for methane before it is released from the aquatic system into the atmosphere."*

L95 "as modified from (Caspers, 1959)." Please refer to how to cite literature and use parentheses (same comment at many places in the MS)
Changed to "modified from Caspers (1959)." and throughout the MS. There are no specific *instructions how to cite when the authors are part of the sentence. I have now changed all respective references to " …. by Smith et al. (2011)*

L107 "were calculated by differential weighing" >I suggest: "were determined gravimetrically"
*Changed accordingly*

L214: change "square" to "rectangle"
*Changed accordingly*

L234: "warm freshwater at the surface (0–5 m), followed by a mixed water body" not clear what "followed" means here… below? Above?
L235 "we found cold and saline water (= "polar water")" please make a sentence
*Changed to ". In September 2013, we observed a sharp stratification, with a warm freshwater layer at the surface (0–5 m) and a mixed water layer immediately below that. Water at depths greater than approximately10 m consisted of cold and saline water (= polar water)."*

L238 "the subsequent stations" unclear, please specify
*Changed to "the stations farther off shore were characterised by polar waters"*

L248 "When applying our water masses (riverine, mixed and polar), we observed…" awkward sentence
*Changed to "We observed significantly different methane concentrations in the riverine, mixed and polar water masses, with medians of 22, 19 and 26 nmol L-1, respectively (p = 0.03;Tab.1). Therefore, we conducted separate linear correlation analyses for each water mass."*

L257 ""riverine" sample) and excluding the very high water values of station TIII---1304; which made the correlation much stronger" Ambiguous use of quotation marks and semicolon. What made the correlation stronger, the high value or to exclude the high value?

*Changed to 2 sentences "With this two modifications, the correlation was much stronger (r2 = 0.62, n= 33, **Error! Reference source not found.**).*
*"*

L272 "The fractional turnover (k') is a measure of the relative activity of the MOBs" > how was k' calculated?

*In the M&M section (L125) we explain "The principle of the MOX estimation is the comparison between the total amount of radioactivity added to the water sample and the radioactive water that was produced due to oxidation of the tritiated methane. The ratio between these values, corrected for the incubation time, is the fractional turnover rate (k'; d-1)"*

L279 "The abundance of MOB can either be given either in cell numbers or as relative abundance" awkward sentence

*Changed to "The abundance of MOBs can be expressed as cell numbers or as relative abundance"*

L290 "Additionally, the "estimated diversity", as OTUs per station," ambiguous use of quotation marks.

*The quotation marks have been removed and the sentence changed to "Additionally, the estimated diversity (OTUs per station), showed a weak but significant …."*

L334 "reported a range of 10 to 100 nmol L---1 (estimated from Figure 2 in (Sapart et al., 2017))." inappropriate use of parentheses (also at many other places in the MS)

*corrected*

L339: "A more detailed comparison with temperate and tropical environments is discussed below, in the context of the diffusive methane flux, as most reviews rely on the methane emissions rather than on the concentrations." What reviews? Please add references

*Changed accordingly*

L345 "seafloor methanogenesis resulting from the decomposition of organic carbon" this is a truism as methanogenesis IS decomposition of OC

*Changed to "In the shallow Chucki Sea, the most likely source was also methane resulting from the decomposition of organic carbon at the seafloor "*

L347 "However, low tides, low topographic relief and low precipitation in the present study area are not favourable for a high ground water input to the Lena Delta." Low tides: do you mean low tidal amplitude? In addition explain the scientific reason for this statements.

*Changed to tidal amplitude*
*In this section, we discuss possible source of methane in the polar /mostly bottom water. Thus, we discuss (beside sediment borne methane) other possible methane sources, which are known for artic seas.*

L353 "Thus, we cannot support ebullition as a methane source." awkward sentence, please check

*Changed to "so ebullition is unlikely to be a significant source of methane. "*

L365 "However, for the latter two processes, it not clear yet if this methane production will result in elevated methane concentrations in situ." Please rephrase
*Changed to "However, the contributions of photosynthesis and DMSP production to in situ methane concentrations remain to be established."*

L368 "However, at the shallow stations (< 8 m, coastal stations of the transects), the water column was mixed; thus, sedimentary methane may diffuse into the water above." Unclear, and oversimplified statement, please rephrase.
*Changed to "However, at the shallower stations (< 8 m, i.e. the coastal stations of the transects), where the water column was mixed, sediments may be the source of the surface water methane."*

L372 "as rivers or estuaries are thought to import methane---rich water into coastal seas." Do you mean "export" instead of "import"?
*Sorry, yes it was "export"*

L378 "However, even this scheme does not seem applicable to our data." Please improve formulation to be more explicit
*Changed to "However, none of the currently proposed schemes seems applicable to our data"*

L393: "median methane oxidation rate of 0.32 nmol L---1 d---1, ranging from 0.03 to 5.7." please avoid numbers without unit
*Unit is now added.*

L401 "The first order rate constant used for modelling the methane flux in the Laptev Sea is estimated to range from 18116 d---1 to 11 d---1 (= 2.3 × 10---6 to 3.8 × 10---3 h---1)" Confusing: was this rate "estimated" or "used"; avoid "=" in the text.
*Changed to "The first order rate constant used for modelling the methane flux in the Laptev Sea ranged from 18116 d-1 to 11 d"*

L407 "The influence of the methane concentration on the MOX was also most pronounced in "riverine water"" inappropriate use of quotation marks
*All quotation marks for "riverine" water (and the others) are now removed.*

L409 "With the described method of qPCR and the water column specific primers (Tavormina et al., 2008), the relative abundance of MOB in our study ranged from 0.05 to 0.47% (median 0.16%) which is equivalent to 4 × 104 to 3 × 106 cells per L (median of 6.3 × 104), except for the high values from station T1---1302." Confusing. What is the high value ?
*We decided to put this sentence to the result section, as we did not want to discuss the outlieres. We also changed the wording to: The relative abundance ranged from 0.05 to 0.47%, except for the high values from station T1-1302, at 1.69 and 2.63% (surface and bottom, respectively, Fig. 6). These high values could not be explained by any environmental or methane-related parameters. In addition, they were statistical outliers and were excluded from further analysis."*

L412: "; thus, they are regarded as methodological outliers" awkward sentence, please justify how such methodological problem may occur.
*They were statistical outliers, the definition is now given in the M&M section, 2.8*

L422 "correlations between parameters important to heterotrophic bacteria," please

rephrase, bacteria don't mind about "parameters"
*Changed to "but we found correlations with parameters that are important for establishment of a heterotrophic bacterial population"*

L451 "Thus, the ecological traits can be described as follows:" not sure "thus" is appropriate at the beginning of a new paragraph. I suggest changing "described" to "summarized"
*Changed to "The ecological traits determined in the present study can be summarised as follows"*

L454 "The relative abundance and "estimated diversity" (OTU/sample) of MOB" inappropriate use of quotation marks
*removed*

L456 "the MOB in the polar population were quite efficient at reaching a MOX comparable to riverine water." Do you mean MOB population in polar waters? do you mean MOX activity, or MOX rates? do you mean comparable to MOX activity of MOB in riverine waters? Please rephrase
*Changed to "The MOBs in the polar population were lower in relative abundance and had a lower estimated diversity than the MOBs in the riverine population, but these microorganisms were quite efficient at reaching a MOX comparable to that observed in riverine water"*

L469 "the classical concept of the r---and k---strategist has today been replaced by the C---S--- R functional classification framework" please explain
*Changed to "replaced by the competitor - stress tolerator - ruderal functional classification framework (Ho et al., 2013)."*

L491 "ranging from 4–163 µmol m2 d---1." Ranging from 4 to 163 µmol m2 d---1
*corrected*

L494 "A comprehensive study by (Myhre et al., 2016)" inappropriate use of parentheses
*corrected*

L498 "(Graves et al., 2015) (Table 5Table 5)." Please edit your MS
*corrected*

L509 "A comprehensive study by (Myhre et al., 2016)" inappropriate use of parentheses
*corrected*

L509 "aquatic methane emissions is presented by (Stanley et al., 2016) and (Ortiz---Llorente and Alvarez---Cobelas, 2012). inappropriate use of parentheses. Same in many other places in the MS
*corrected*

L512 "as well as from tidal systems, to which the Lena Delta would be classified" "to which" is not correct
*Changed to "This finding contrasts with the review by Borges and Abril (2012) comparing worldwide estuaries, where an increase in methane emissions was evident from estuaries at high latitudes, as well as from tidal systems. (Notably, the Lena delta matches both of these classifications.)"*

L514 "; thus they conclude" to what do "they" refer to ?
*To the authors Ortiz-Llorente and Alvarez-Cobelas, changed to "the authors"*

L520 and at many other places in the MS "(Borges et al., 2017;Lofton et al., 2014)" insert a space after the semicolon
*Corrected in the whole Ms*

L525 "A molecular approach identified the salinity, temperature and pH as the most important environmental drivers of methanogenic community composition on a global scale." Please add a reference
*corrected*

L534 a paragraph cannot start with ")."
*corrected*

L534 "he methane released by ebullition did not show any isotopic evidence of oxidation;" please provide a reference
*The reference is at the end of the sentence (Sapart et al 2017).*

L556 "However, one fact to be kept in mind is that our estimation is a static one" poor English
*Changed to "Our estimate of methane flux is a static one and does not take into account the currents and spreading of the freshwater plume."*

L561 "A more complex approach is taken by (Wahlström and Meier, 2014)." Poor English and inappropriate use of parentheses
*Changed to "The simulations performed by Wahlström and Meier (2014) revealed the importance of the methane oxidation rate constant and the crucial necessity of obtaining an in situ measurement of it"*

L576 "as no direct dilution of riverine methane occurs;" awkward sentence, how can "dilution" be "direct" (or "indirect")?
*Changed to "as we did not find evidence of a direct methane input of the Lena River"*

L583 "; thus, we propose that it will compensate for any increase in methane concentrations." Not clear, is this speculation? Do you mean you "postulate"?
*Yes, corrected to "postulate"*

L585 "he stratification of the water column will be broken up and the separate water masses mixed" looks like spoken English, not written English.
*Changed to "However, increases in storm frequency or strength will disrupt the stratification of the water column and promote mixing of the different water masses"*